# SolverLLM: Leveraging Test-Time Scaling for Optimization Problem via LLM-Guided Search

**Dong Li**[1]    **Xujiang Zhao**[2]*    **Linlin Yu**[3]    **Yanchi Liu**[2]    **Wei Cheng**[2]
**Zhengzhang Chen**[2]    **Zhong Chen**[4]    **Feng Chen**[5]    **Chen Zhao**[1]*    **Haifeng Chen**[2]

[1]Baylor University    [2]NEC Labs America    [3]Augusta University
[4]Southern Illinois University    [5]University of Texas at Dallas
{dong_li1, chen_zhao}@baylor.edu   {xuzhao, Haifeng}@nec-labs.com

## Abstract

Large Language Models (LLMs) offer promising capabilities for tackling complex reasoning tasks, including optimization problems. However, existing methods either rely on prompt engineering, which leads to poor generalization across problem types, or require costly supervised training. We introduce SolverLLM, a training-free framework that leverages test-time scaling to solve diverse optimization problems. Rather than solving directly, SolverLLM generates mathematical formulations and translates them into solver-ready code, guided by a novel Monte Carlo Tree Search (MCTS) strategy. To enhance the search process, we modify classical MCTS with (1) dynamic expansion for adaptive formulation generation, (2) prompt backpropagation to guide exploration via outcome-driven feedback, and (3) uncertainty backpropagation to incorporate reward reliability into decision-making. Experiments on six standard benchmark datasets demonstrate that SolverLLM outperforms both prompt-based and learning-based baselines, achieving strong generalization without additional training.

## 1   Introduction

An optimization problem seeks the best possible decision in terms of a numeric objective while satisfying a set of specified constraints. Such problems ground decision-making in engineering [4], energy management [14], economics [8], healthcare [7], and many other areas [21]. Solving an optimization problem typically involves three stages. *Problem Formulation*, translate the problem from a domain-specific, often natural language description into a precise mathematical formulation specifying the variables, constraints, and objective function. *Code Generation*, translate the mathematical model into executable code. *Program Execution*, run the code with a standard optimization solvers such as Gurobi or Pyomo [9, 6]. Among these steps, problem formulation demands expertise in both the application domain and mathematical programming, limiting broader adoption and automation of optimization-based decision-making.

The recent and rapid development of Large Language Models (LLMs) have ushered in a new era of capabilities in complex reasoning and natural language understanding. Moreover, combining LLMs with algorithmic components yields competitive and reliable task performance with moderate computation cost [25, 29, 18]. In optimization problems, an LLM can generate the mathematical formulation and corresponding code, while a proven solver executes this code to obtain a reliable solution. The Natural Language for Optimization (NL4Opt) benchmark [19] captures this setting by requiring models to convert a textual description into a formal program. Existing solutions can be roughly divided into two categories. Prompt-based methods [23, 2] coordinate specialized agents

---

*Chen Zhao and Xujiang Zhao are corresponding authors.

under a carefully designed workflow, which makes them sensitive to prompt choices. Learning-based methods [17, 22, 12] fine-tune a general LLM on curated problem–solution pairs, but their effectiveness depends on significant dataset labeling and model fine-tuning cost.

Motivated by the effectiveness of test-time scaling techniques, which allocate additional computation during inference to boost task performance without extra training cost [27], we propose SolverLLM, a training-free framework that leverages test-time scaling to solve diverse optimization problems. Rather than predicting solutions directly through multi-agent prompting or task-specific fine-tuning, SolverLLM decomposes problem formulation into essential stages and explores formulation space with multiple call of LLM guided by a Monte Carlo Tree Search (MCTS) strategy. Figure 1 presents a comparison of SolverLLM with prompt-based and learning-based methods. Our contributions are threefold:

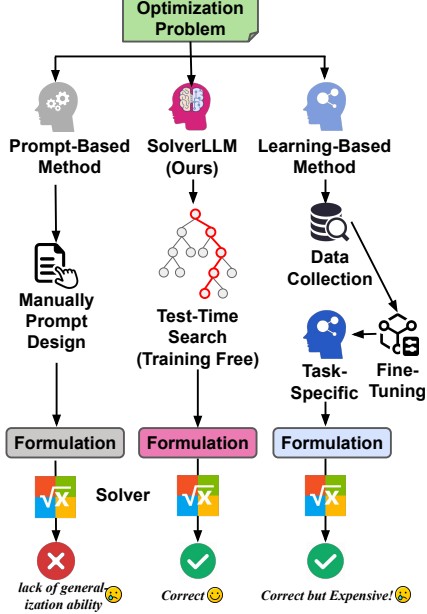

Figure 1: Comparison of the solution pipelines employed by prompt-based approaches, learning-based approaches, and SolverLLM for optimization problems.

- We proposed **SolverLLM**, a training-free framework that leverages a test-time scaling strategy to solve diverse optimization problems.

- SolverLLM introduces three key innovations: (i) *dynamic expansion*, which lets the LLM incrementally add or refine variables and constraints; (ii) *prompt backpropagation*, which feeds solver feedback through the tree to steer subsequent prompt edits; and (iii) *uncertainty backpropagation*, which incorporates reward variance to improve search efficiency.

- Extensive experiments on six standard benchmark datasets show that SolverLLM consistently outperforms leading prompt-based and learning-based baselines, achieving a 10% improvement over the state-of-the-art.

## 2 Related Work

**Prompt-Based Methods for Optimization Problems**. Chain-of-Experts [23] tackles the automated formulation of optimization problems as a dialogue among specialized LLM agents. A conductor invokes an interpreter, modeller, coder, and reviewer in sequence, then walks back along the chain so each agent can reflect on and revise its own output. The loop repeats until the generated python code solves the instance. OptiMUS [2] adopts a similar multi-agent approach but changes the workflow. It first converts the problem description into a structured record of parameters, objectives, constraints, and context. A manager agent then cycles through a formulator, a programmer, and an evaluator to express each clause mathematically, generate code, run the program, and correct errors as needed. Although effective, these methods rely on carefully tuned agent roles and prompt templates, making them fragile on unfamiliar optimization tasks.

**Learning-Based Methods for Optimization Problems**. ORLM [22] introduces a semi-automatic data-synthesis framework that iteratively expands and augments a small real-world dataset. The model is fine-tuned to take a textual description as input and to produce both the mathematical formulation and the corresponding solver implementation. LLMOPT [12] assembles expert-verified samples covering five components(sets, parameters,objective, variables, constraints) samples with GPT-4 assistance. The model uses supervised fine-tuning and value-based alignment to reduce hallucinated outputs, and it iteratively refines solutions during inference through a solver-guided self-correction loop. These approaches rely on large, high-quality optimization datasets, and their ability to handle unseen problem families and the substantial computational cost of training remains open concerns.

**Test-Time Scaling with LLMs**. Test time scaling allocates extra computation only during inference to extract more reasoning power from a frozen language model [27]. Typical strategies include repeated decoding passes [5], extended chains of thought [26], and prompt-space search [24]. Notably, search

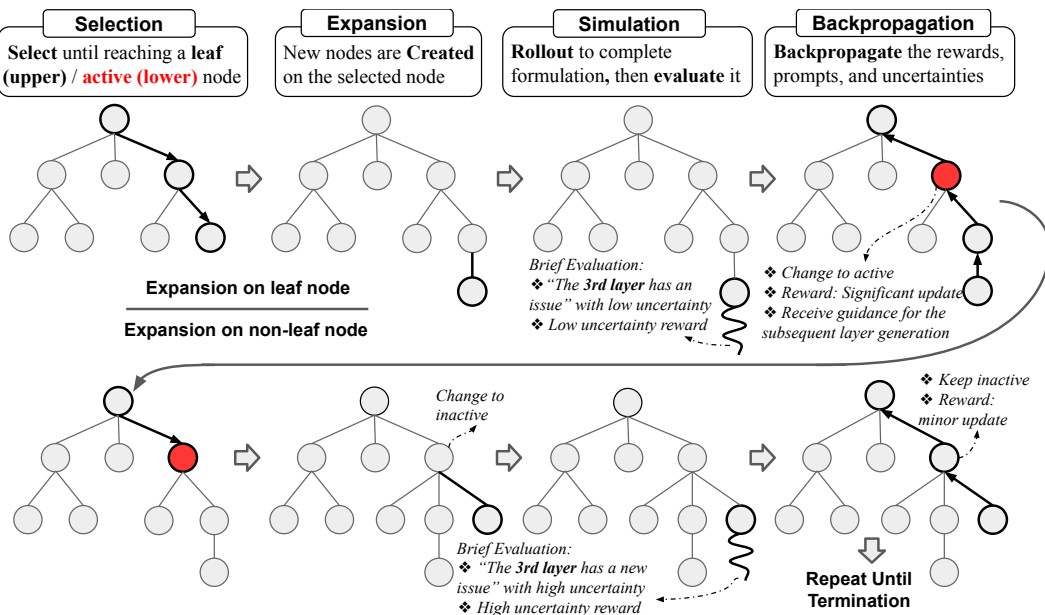

Figure 2: An illustration of SolverLLM with two MCTS iterations, each consisting of four stages: **selection**, **expansion**, **evaluation**, and **backpropagation**. Unlike standard MCTS, which performs expansion only on leaf nodes (as in the first iteration), SolverLLM also enables dynamic expansion at non-leaf nodes (as in the second iteration). Beyond rewards, the evaluation phase additionally generates reasoning signals that provide layer-specific guidance via backpropagation for subsequent search steps. Furthermore, this phase estimates both global and local uncertainties for rewards and reasoning signals, which are leveraged in the backpropagation phase to accelerate the search process.

techniques like MCTS demonstrates significant performance increases over math and reasoning tasks [28, 16]. Our work builds upon this line by incorporating a structured MCTS framework, augmented with LLM-driven feedback and uncertainty estimation, to navigate the formulation space of optimization problems at inference. This training-free paradigm enables SolverLLM to generalize across domains while avoiding the limitations of both prompt sensitivity and costly training.

## 3 Methodology

### 3.1 General Form of Optimization Problems

Optimization problems aim to find the best outcome under given constraints by minimizing or maximizing an objective function over decision variables. The standard form is:

$$\min_{\mathbf{x} \in \mathcal{X}} \quad f(\mathbf{x}) \text{ s.t.} \qquad h_i(\mathbf{x}) \leq 0, \quad i = 1, \dots, m \tag{1}$$

Here, $\mathbf{x} \in \mathbb{R}^D$ is the $D$-dimensional decision variable, $\mathcal{X}$ the feasible set, $f$ the objective function, and $h_i$ the constraints. SolverLLM builds upon this abstraction by treating the formulation process itself as a search problem, enabling structured exploration of $f$, $\mathcal{X}$, and the constraints through language model generation.

### 3.2 SolverLLM: Optimization Formulation via LLM-Guided MCTS

To tackle the complexity of formulating diverse optimization problems from natural language, we introduce **SolverLLM**, a training-free framework that leverages a Monte Carlo Tree Search (MCTS) algorithm guided by LLMs. SolverLLM treats the formulation process itself as a structured decision-making problem, where each step incrementally builds or refines a formulation based on LLM-generated proposals and solver feedback. The overview of SolverLLM is shown in Figure 2. SolverLLM builds this tree through four canonical MCTS phases—**selection**, **dynamic expansion**, **simulation**, and **backpropagation**—each adapted to support symbolic reasoning with language

models. Before describing these components, we introduce our element-based decomposition that guides the search.

### 3.2.1 Element-Based Formulation as Search Guidance

Inspired by five-element abstraction from prior work [12], we design a six-element schema: Type, Sets, Parameters, Variables, Objective, and Constraints. These elements guide LLM-driven expansion and enable structured, interpretable search via MCTS.

A key enhancement in SolverLLM is the introduction of the **Type element**, which identifies the high-level category of the optimization problem, such as linear programming, or integer programming. This early-stage classification provides a form of global guidance $G_g$ before the LLM begins constructing detailed formulations. Analogous to a student reviewing key concepts before solving exam problems, the Type element helps the model establish a coherent mental model of the task at hand. This early guidance helps the LLM form a coherent task model, reducing ambiguity and ensuring consistency in later decisions, especially for complex or unfamiliar problems. SolverLLM constructs formulations incrementally across these six elements. Each node encodes a partial formulation, while the entire path of nodes defines the complete model. This modular design enables semantic consistency, local reasoning, and flexible correction, making it ideal for test-time inference.

### 3.2.2 Selection

The selection phase in SolverLLM traverses the current search tree from the root to either a promising leaf node eligible for expansion or an active non-leaf node. At each step, we select one of the current node's children by balancing exploitation (high-reward nodes) and exploration (less-visited nodes). SolverLLM adopts the Upper Confidence Bound for Trees (UCT) for this purpose. During the entire search process, each node $s$ continuously maintains two statistics based on solver feedback collected from past rollouts: a total visit count $N_s$ and an average reward $Q_s$. Given a parent node $s$, the next child $s_{child} \in \text{Child}(s)$ is selected according to:

$$s_{child} = \arg \max_{s' \in \text{Child}(s)} \left[ Q_{s'} + c \cdot \sqrt{\frac{2 \log N_s}{N_{s'}}} \right],$$

where $c$ is an exploration constant controlling the degree of exploration.

Unlike standard MCTS, where each node represents a complete state and the selection phase terminates only upon reaching a leaf node, each node $s$ in SolverLLM corresponds to a partially constructed optimization formulation represented as a subset of the six elements introduced in Section 3.2.1, and the selection process can also terminate early when an active non-leaf node determined by a trigger $t_s$ (discussed in Section 3.2.5) is encountered, enabling dynamic expansion.

### 3.2.3 Dynamic Expansion

At the heart of SolverLLM's dynamic expansion strategy is the use of a language model to generate new formulation candidates in an open-ended and context-aware manner. Unlike traditional MCTS, where the action space is predefined and finite, the space of valid optimization formulations is vast and unstructured. Rather than relying on a static set of predefined actions [3], SolverLLM prompts the LLM to produce new child nodes tailored to the current partial formulation, which increases the breadth of the search and facilitates solving more complex optimization problems. The dynamic expansion strategy of SolverLLM mainly consists of the following two core components:

**Expansion on non-leaf nodes.** Benefiting from the modified selection strategy, expansion can be performed on non-leaf nodes. This flexibility accounts for the non-linear structure of optimization problem formulation: elements such as variables and constraints may depend on or revise earlier components. SolverLLM can revisit and refine earlier decisions based on updated feedback, enabling deeper and more accurate formulations over time.

**LLM-guided expansion with local reasoning.** In SolverLLM, each element layer $l$ is assigned a local expert-guided knowledge base $\mathcal{G}_l$ that guides the expansion of any node (whether it's leaf or non-leaf nodes) within that layer. This knowledge base is constructed from accumulated reasoning signals derived from the evaluation of past generated formulations, which will be discussed in detail in Section 3.2.5. Such formulation-as-feedback loop allows SolverLLM to continuously adapt its expansion behavior, effectively learning from past mistakes during inference.

Overall, this dynamic, reasoning-aware expansion process equips SolverLLM with the ability to construct complex, high-quality formulations through iterative refinement and signals feedback.

### 3.2.4 Simulation

The simulation phase comprises two key operations performed on the expanded node $s$: **rollout** and **evaluation**. Rollout refers to the process of simulating a complete solution from a specific node by iteratively applying actions until a terminal state or fully constructed formulation is reached. Once the complete formulation is obtained, SolverLLM evaluates its quality by translating it into executable code and running it through a solver. This step determines whether the formulation is syntactically valid, solver-compatible, and capable of producing a feasible or optimal solution. The purpose of the evaluation is to obtain the following signals, which serve to guide the subsequent search process:

**Reward.** Each formulation $f_s$ at node $s$ is translated into code and solved by a numerical solver, yielding a output $x^*$. Subsequently, a reward $R(f_s, x^*)$ is assigned, which captures several factors:

- **Feasibility:** Whether the solver ran successfully and found a feasible solution.
- **Optimality:** Whether the solution meets the problem's objective (e.g., minimizes cost).
- **Error Penalty:** Whether the code execution failed or returned invalid outputs.

The reward is computed as a weighted sum:

$$R(f_s, x^*) = \alpha \cdot \mathbb{I}_{\text{feasible}} + \beta \cdot \texttt{objective\_score}(f_s, x^*) - \gamma \cdot \mathbb{I}_{\text{error}}$$

where $\alpha$, $\beta$, and $\gamma$ are hyperparameters, and $\mathbb{I}_{\text{feasible}}$, $\mathbb{I}_{\text{error}}$ are binary indicators. The term $\texttt{objective\_score}(f_s, x^*)$ represents a subjective judgment of the solution quality, based on both the original formulation and the computed result. Specifically, we prompt the LLM to act as a lightweight evaluator (or "judger") that assesses how well the solution aligns with the intent and structure of the formulation $f_s$. This allows us to estimate relative solution quality even in the absence of exact ground-truth labels or reference objectives, and is particularly useful in cases involving heuristic or approximate solvers.

**Reasoning Signals of Each Layer.** Beyond rewards, we further employ the LLM to evaluate each element in the generated formulation, producing corresponding reasoning signals for each layer. Specifically, for each layer $l$ and its associated node $s_l$ in the formulation, we define the reasoning signals as a triplet $\mathcal{S}_l = (t_{s_l}, E_{s_l}, G_l)$, where $t_s$ denotes a one-time trigger indicating node activation status of node $s_l$, $E_{s_l}$ represents the reason explaining whether node $s_l$ is appropriate, and $G_l$ provides layer-level prompt guidance for revision when the node $s_l$ is deemed inappropriate. These signals are propagated backward, serving as critical information that influences the subsequent search process of SolverLLM. A detailed description of the reward and reasoning signals is provided in Appendix C.2.

### 3.2.5 Backpropagation

After evaluating a candidate formulation and obtaining its reward and reasoning signals, SolverLLM performs backpropagation to update the search tree and inform future decisions. In standard MCTS, this step updates visit counts and value function along the path from the current node back to the root. SolverLLM extends this process with two key innovations tailored for language-model-guided reasoning: prompt backpropagation and uncertainty propagation.

**Prompt Backpropagation.** Traditional MCTS propagates only scalar rewards, thereby overlooking rich contextual feedback, especially when the formulation itself is imperfect. In SolverLLM, the reasoning signals $\mathcal{S}_l$ for each layer $l$ obtained from evaluation are treated as feedback. Specifically, for each node $s$ at each level $l$ along the path used to generate the formulation, we propagate the corresponding reasoning signals $\mathcal{S}_l$ backward, updating the node's state until reaching the root. If a trigger $t_{s_l}$ is present in $\mathcal{S}_l$, the corresponding node $s$ is regarded as activated, meaning it becomes eligible for further expansion, consistent with Section 3.2.2. In addition, we construct a knowledge base $\mathcal{G}_l$ for each layer $l$ to continuously accumulate prompt guidance $G_l$ from $\mathcal{S}_l$. This guidance is then incorporated into future prompt construction during dynamic expansion, enabling reasoning-aware search and formulation refinement. However, given the uncertainty associated with LLM outputs, we additionally compute local uncertainty $U_{s_l}^{\text{local}}$ based on $E_{s_l}$, using predictive entropy [13]:

$$U_{s_l}^{\text{local}} = \mathbb{E}_{E_{s_l}} \left[ \frac{1}{|E_{s_l}|} \sum_{a_i}^{E_{s_l}} - \log \mathbb{P} \left( a_i \mid a_0, \cdots a_{i-1} \right) \right],$$

where $a_i$ is the $i$-th token of $E_{s_l}$. Only when $U_{s_l}^{\text{local}}$ exceeds a threshold $\eta$ and a trigger $t_{s_l}$ is present will we activate the corresponding node $s$ and perform prompt backpropagation.

**Uncertainty Backpropagation.** Leveraging an LLM as a semantic scorer enables task-agnostic reward estimation, but introduces high variance due to the subjectivity and variability of LLM outputs. This makes reward propagation unstable, especially for complex or ambiguous formulations. To mitigate this, we incorporate reward uncertainty as global uncertainty during backpropagation. Specifically, we estimate the semantic uncertainty [15] at each evaluated node $s$ by repeatedly sampling `objective_score`$(f_s, x^*)$, yielding an uncertainty measure $U_s^{\text{global}}$, as detailed in Appendix B. We then down-weight the impact of uncertain evaluations during backpropagation. For each node $s'$ on the path from $s$ to the root, we update its value using an uncertainty-weighted average:

$$Q_{s'} \leftarrow Q_{s'} + \rho_s \cdot \frac{\bar{R}(f_s, x^*) - Q_{s'}}{N_{s'}}$$

where the trade-off factor is defined as a decreasing function of uncertainty: $\rho_s = \exp(-U_s^{\text{global}})$, and $\bar{R}(f_s, x^*)$ is the average reward based on multiple samplings. This formulation ensures that confident evaluations (low variance) are strongly propagated, while noisy or uncertain judgments have limited influence on tree statistics. Finally, for every node $s'$ on the path, we also increment the visit count: $N_{s'} \leftarrow N_{s'} + 1$. These updates to $Q_{s'}$ and $N_{s'}$ are used in the next selection step to determine which parts of the tree should be further explored.

By combining symbolic reasoning feedback (prompt backpropagation) and statistical robustness (uncertainty backpropagation), SolverLLM transforms MCTS from a purely numerical search into a feedback-rich, language-informed inference process. This enables more intelligent reuse of partial formulations, better adaptation to errors, and faster convergence to high-quality solutions.

## 4 Experiments

To evaluate the effectiveness of **SolverLLM**, we conduct comprehensive experiments across six benchmark datasets, comparing our approach with both prompt-based and learning-based baselines. Our study is designed to answer the following key research questions:

- **(Q1) SolverLLM vs. Baseline Methods:** How does SolverLLM, as a test-time scalable framework, compare to prompt-based methods? Can SolverLLM match or exceed the performance of training-intensive approaches without incurring the cost of data collection and fine-tuning?

- **(Q2) Impact of Dynamic Expansion:** What role does the dynamic formulation expansion, enhanced by prompt backpropagation, play in improving solution accuracy?

- **(Q3) Effectiveness of Uncertainty Backpropagation:** How does incorporating uncertainty estimates into the search process improve efficiency and decision quality during inference?

- **(Q4) Importance of Type Element:** How does the inclusion of type element in the six-element formulation affect its effectiveness compared to the widely explored five-element one?

These questions guide the structure of our experimental analysis in the following sections, which include detailed setup, comparative evaluation, and ablation studies.

### 4.1 Experimental Setup

For evaluation, we use the test set portions from six real-world optimization and operation task datasets: NL4Opt [19], Mamo (EasyLP and ComplexLP) [10], NLP4LP [2], ComplexOR [23], and IndustryOR [22]. These datasets include optimization problem cases of varying difficulty, types, and domains. Among them, the test set of NLP4LP is obtained by shuffling the source dataset and randomly sampling 100 cases. All other datasets use the same setting as LLMOPT [12]. We evaluate the effectiveness of the methods on the optimization problems using *Solving Accuracy (SA)* and

*Execution Rate (ER)*. SA is the proportion of optimization problem cases successfully solved by the algorithm. ER is the proportion of cases where the code runs successfully without any errors and produces output. Additionally, we use *Average Generation Times (AGT)* (in minutes) to measure the efficiency of the methods, which refers to the time required for the method to model the formulation of the problem. Detailed experimental information are provided in Appendix A.

## 4.2 Result Analysis

We selected GPT-4 [1] and GPT-4o [11], which are used directly, along with prompt-based methods Reflexion [20], Chain-of-experts [23], OptiMUS [2], test-time scaling based method AutoFormulation [3], as well as learning-based methods including ORLM [22] (built on Mistral-7B, Deepseek-Math-7B-Base and LLaMa3-8B), and LLMOPT [12] (built on Qwen1.5-14B) as the compared methods for a comprehensive comparison.

**Comparison with Prompt-Based Methods.**
The experimental results of Solving Accuracy (SA) for the Comparison with prompt-based methods are shown in Table 1. For comparability, we have retained the results from the original paper as much as possible. The performance of SolverLLM significantly outperforms these methods, with improvements exceeding 10% in all datasets. This can be attributed to its test-time scalable framework, which directs the LLM to break down the problem into six elements and leverages MCTS with dynamic expansion to explore a broader set of potential formulations, ultimately selecting the optimal one. This approach significantly improves the model's capability to solve optimization problems more effectively.

Table 1: The comparison results of SA between prompt-based methods and SolverLLM. The baseline results are cited from [12]. **Bold** is the best, while underlined is the second-best.

|  | NL4Opt | NLP4LP | ComplexOR |
|---|---|---|---|
| GPT-4 Directly | 47.3% | 35.8% | 9.5% |
| GPT-4o Directly | 81.0% | 32.4% | 27.3% |
| Reflexion | 53.0% | 46.3% | 19.1% |
| Chain-of-Experts | 64.2% | 53.1% | 38.1% |
| OptiMUS | 78.8% | 72.0% | 66.7% |
| **SolverLLM (Ours)** | **97.0%** | **87.0%** | **77.8%** |

**Comparison with Learning-Based Methods.** We compared SolverLLM with recent optimization-specific LLMs trained via Supervised Fine-Tuning (SFT), as shown in Table 2. As a training-free method, Solver-LLM significantly outperforms ORLM on SA and matches the state-of-the-art LLMOPT on simpler datasets like MamoEasy and NL4Opt. On more challenging datasets such as MamoComplex and IndustryOR, it surpasses LLMOPT by 8 and 10 percentage points, respectively. This demonstrates that SolverLLM delivers strong performance without supervised data or fine-tuning overhead. In contrast, learning-based methods are limited by their training data distribution and struggle to generalize. Even after fine-tuning, they often underperform on complex, unseen problems. SolverLLM mitigates this by exploring a broader solution space, leveraging prompt backpropagation to accumulate experience over time—trading off evaluation speed for greater robustness across diverse optimization tasks.

Table 2: The comparison results of SA between learning-based methods and SolverLLM. The baseline results are cited from [12]. **Bold** is the best, while underlined is the second-best.

|  | MamoEasy | NL4Opt | MamoComplex | IndustryOR |
|---|---|---|---|---|
| GPT-4 Directly | 66.5% | 47.3% | 14.6% | 28.0% |
| GPT-4o Directly | 91.0% | 81.0% | 34.0% | 34.0% |
| ORLM-Mistral | 81.4% | 84.4% | 32.0% | 27.0% |
| ORLM-Deepseek | 82.2% | 86.5% | 37.9% | 33.0% |
| ORLM-LLaMa3 | 82.3% | 85.7% | 37.4% | 38.0% |
| LLMOPT | **97.0%** | 93.0% | 68.0% | 46.0% |
| **SolverLLM (Ours)** | 96.0% | **97.0%** | **76.0%** | **56.0%** |

**Comparison with Other Test-Time Scaling Methods.** As a recent test-time scaling method also leveraging Tree of Thought (ToT), AutoFormulation [3] employs MCTS to automatically generate the four-element formulation. We compared the SA and ER results of AutoFormulation and SolverLLM across six datasets, as shown in Figure 1. SolverLLM outperforms AutoFormulation in SA on both simple and complex datasets, indicating that the dynamic expansion enabled by prompt backpropagation leads to a better search tree, which aids in exploring the correct formulation. Furthermore, even with the more detailed six-element formulation, SolverLLM still achieves a higher ER, with almost all the code executing successfully. This is attributed to its use of error backpropagation during the code generation phase, as detailed in Appendix C.

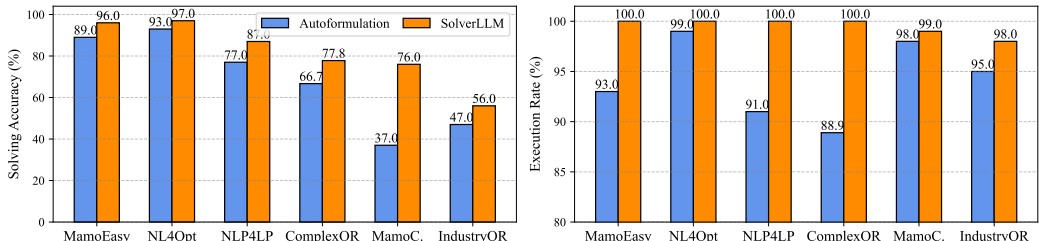

Figure 3: The comparison of SA and ER results between AutoFormulation and SolverLLM (Ours) across six real-world dataset. "MamoC." is the abbreviation for MamoComplex.

Table 3: The ablation study results on three metrics across six datasets for SolverLLM and its variants without (w/o) Prompt Backpropagation (PB), Uncertainty Backpropagation (UB), and Type Element (TE). Bold indicates the best performance. ↑ means higher is better, ↓ means lower is better.

| Datasets (Easy→Hard) | SA (%) ↑ / ER (%) ↑ / AGT (min) ↓ | | | | | |
| | **MamoEasy** | **NL4Opt** | **NLP4LP** | **ComplexOR** | **MamoComplex** | **IndustryOR** |
|---|---|---|---|---|---|---|
| **SolverLLM** | **96.0** / **100.0** / 2.53 | **97.0** / **100.0** / 2.33 | **87.0** / **100.0** / 2.38 | **77.8** / **100.0** / 2.91 | **76.0** / **99.0** / 3.85 | **56.0** / **98.0** / 3.28 |
| w/o PB | 90.0 / 99.0 / **2.32** | 93.0 / **100.0** / **2.19** | 81.0 / **100.0** / **2.13** | 66.7 / 94.4 / 2.67 | 69.0 / **99.0** / 3.81 | 46.0 / 96.0 / 3.24 |
| w/o UB | 95.0 / **100.0** / 2.57 | **97.0** / **100.0** / 2.30 | 85.0 / 99.0 / 2.42 | **77.8** / 94.4 / 3.27 | 75.0 / **99.0** / 4.34 | **56.0** / 97.0 / 3.68 |
| w/o TE | 92.0 / 99.0 / 2.41 | 96.0 / **100.0** / 2.48 | 82.0 / 99.0 / 2.58 | 72.2 / **100.0** / 2.51 | 59.0 / **99.0** / **3.56** | 48.0 / **98.0** / **3.01** |

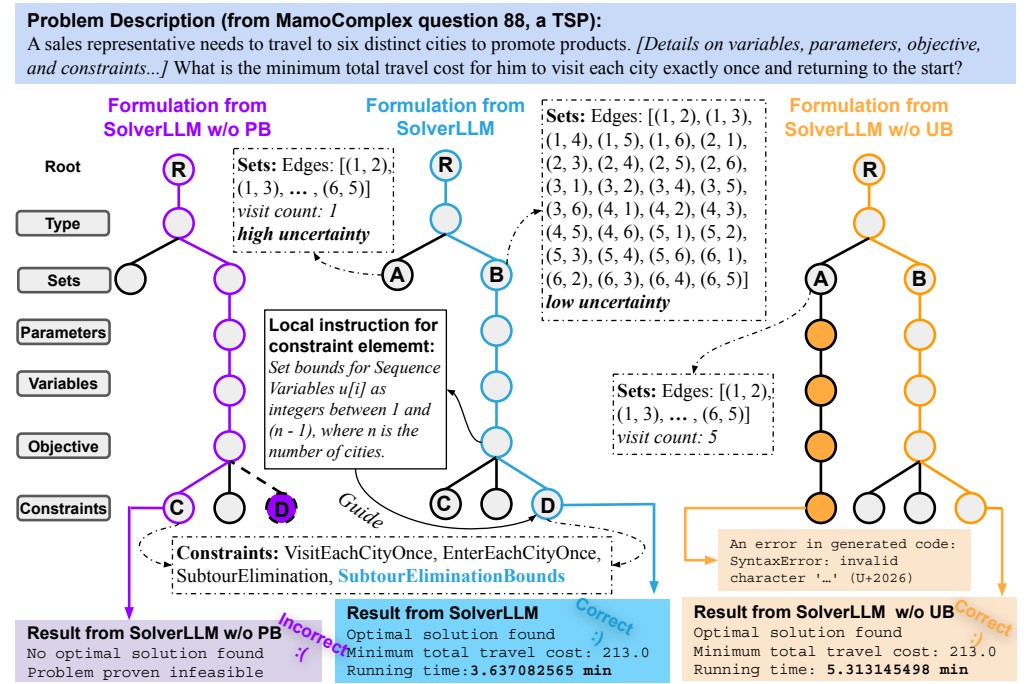

Figure 4: A case study of SolverLLM and its variants w/o PB and w/o UB on a hard TSP optimization problem with their search trees. Node fill colors show changes from SolverLLM; solid lines mean additions, dashed lines mean deletions. The content of nodes A and B (related to the w/o UB variant) and nodes C and D (related to the w/o PB variant) has been indicated by the dotted-dashed line. The local instruction generated by PB are used to guide the generation of the constraints element. The colored paths (violet, blue and orange) represent the optimal formulations, which ultimately point to the results. In this problem, SolverLLM and its w/o UB variant produced correct results, with the former being more efficient.

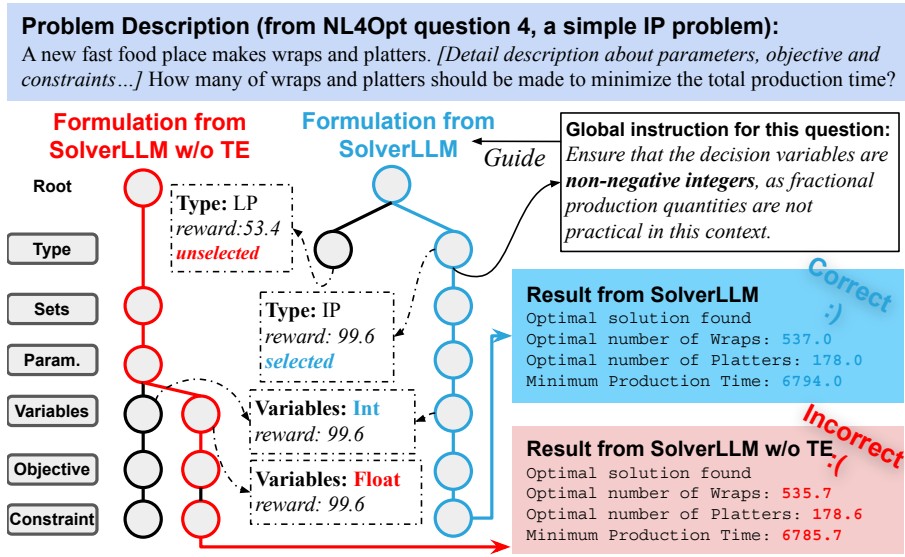

Figure 5: A case study of SolverLLM and its variant w/o TE on a simple IP problem with their search trees. The content of notable nodes has been indicated by the dotted-dashed line. The global instructions generated by TE are used to guide the generation of all elements. The colored paths (red and blue) represent the optimal formulations, which ultimately point to the results.

## 4.3 Ablation Study

To explore the impact of the main components of SolverLLM in solving optimization problems, we conducted a comprehensive ablation study on three variants of SolverLLM: without (w/o) Prompt Backpropagation (PB), without Uncertainty Backpropagation (UB), and without Type Element (TE), with the main results shown in Table 3. We also conducted case studies to gain a deeper understanding of each component by analyzing the search trees, as shown in Figure 4 and Figure 5.

**Impact of Dynamic Expansion.** SolverLLM shows significant performance improvement and better ER compared to SolverLLM w/o PB in Table 3, especially on more challenging datasets. This highlights that dynamic expansion via PB enhances the model's ability to explore optimal formulations. As shown in Figure 4, focusing on nodes C and D, SolverLLM successfully solved the Traveling Salesman Problem (TSP), while SolverLLM w/o PB failed due to missing constraints on the sequence variable range, causing subtour elimination to fail. SolverLLM w/o PB is restricted by its structure and, after receiving two distinct constraint definitions, can no longer explore further, settling for the best result within its current structure. In contrast, dynamic expansion allows SolverLLM to overcome these limitations, enabling bidirectional influence in MCTS exploration—upper-level nodes can influence lower-level ones, and vice versa. Through local instructions generated by PB, the correct constraint definitions at node D were successfully explored.

**Effectiveness of Uncertainty Backpropagation.** As shown in Table 3, UB enhances the model's efficiency while preserving its effectiveness. SolverLLM and SolverLLM w/o UB achieved similar SA and ER results. While their AGT is comparable on simpler datasets, SolverLLM is significantly more efficient on the more complex ones. This is due to UB's ability to prune low-quality nodes earlier, simplifying the search tree and enabling faster exploration. In the TSP example in Figure 4, nodes A and B model similar sets. They have similar initial rewards, but the set in node A is incomplete, which makes the formulation modeled based on this node likely to generate incorrect code, making it unreliable. SolverLLM w/o UB fully explored node A's branch before identifying the issue, wasting a significant amount of time, even though the correct result was obtained. In contrast, SolverLLM avoided further expansion of node A's subsequent nodes early on through UB, enabling it to reach the correct result with a smaller time cost.

**Importance of Type Element.** With the help of TE, SolverLLM achieves modest improvements on simple datasets and significant gains on more challenging ones. This is because TE is designed to provide the model with global instructions about the current problem, enabling it to stay focused

on critical details during formulation. This improves modeling completeness for complex problems. Moreover, on relatively simple datasets, TE sometimes even reduces time consumption, as it can contribute to pruning in simpler search tree structures. We further illustrate the role of TE using a simple Integer Programming (IP) problem, as shown in Figure 5. SolverLLM w/o TE possibly incorrectly modeled the variable as a floating-point type, leading to incorrect results. In contrast, SolverLLM avoided this mistake under the guidance of the type element and the global instruction it generates. We also observed that TE is particularly effective for graph-related problems. To further investigate this, we conducted a case study on such a problem, as detailed in Appendix B.

## 4.4   Token Budget Analysis

Token usage is a critical factor for test-time scaling methods, as it directly determines inference cost and practical deployability. To assess token efficiency, we compare SolverLLM with AutoFormulation on the MamoComplex dataset under identical configurations, varying the maximum number of search iterations from 10 to 50. For each budget, we record the total number of tokens consumed during inference and the corresponding solving accuracy (SA), ensuring all runs share identical hardware and decoding settings for fair comparison. As summarized in

Table 4: Token consumption and SA under different search iterations on MamoComplex.

| | SolverLLM | | AutoFormulation | |
|---|---|---|---|---|
| # Iter | # Tokens | SA (%) | # Tokens | SA (%) |
| 10 | 32,790 | 69.0 | 35,911 | 34.0 |
| 20 | 40,920 | 76.0 | 43,150 | 37.0 |
| 30 | 49,337 | 77.0 | 54,427 | 38.0 |
| 40 | 57,834 | 78.0 | 62,248 | 38.0 |
| 50 | 66,312 | 78.0 | 70,245 | 40.0 |

Table 4, SolverLLM consistently achieves higher solving accuracy while consuming fewer tokens across all search budgets. This advantage stems from its reasoning-aware search design, where prompt and uncertainty backpropagation jointly guide exploration toward semantically meaningful formulation paths, effectively avoiding redundant or low-quality expansions.

## 5   Conclusion

We introduced SolverLLM, a training-free framework that leverages LLM-guided Monte Carlo Tree Search to solve diverse optimization problems at test time. Unlike prompt-based or fine-tuned methods, SolverLLM incrementally generates, evaluates, and refines formulations without task-specific training. Our method features dynamic expansion for incremental formulation construction, prompt backpropagation for feedback-driven refinement, and uncertainty propagation to improve search robustness. Experiments on six benchmarks show SolverLLM outperforms both prompt- and learning-based baselines, with ablations confirming the contribution of each component. This work highlights the potential of test-time reasoning in structured domains and opens avenues for extending to more complex optimization settings and hybrid inference paradigms.

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

# A Experimental Details

## A.1 Detailed Datasets

We use the test set portions from six real-world optimization and operation task datasets: NL4Opt [19], Mamo (EasyLP and ComplexLP) [10], NLP4LP [2], ComplexOR [23], and IndustryOR [22], which include optimization problem cases of varying difficulty, types, and domains. For a deeper understanding of the dataset, we designed the following prompt to classify the difficulty level of each optimization problem in the dataset using GPT-4o.

---

**Prompt for determining the difficulty level of optimization problems**

```
Given an optimization problem, your task is to determine the
**difficulty** of the optimization problem.

Here's the problem description:
[problem_description]

Please judge the type of difficulty optimization problem.  Return your
response as a Python list with a single dictionary.
* The dictionary should have a 'difficulty' key, which indicates the
difficulty of this problem.

The difficulty have five levels:
LEVEL 1:  Simple Linear Optimization Problem
- Features:  The mathematical expression is explicitly stated,
requiring no translation from text into a mathematical model; very
few variables (<=2), and constraints are simple and direct.
- Typical Problem Example:  "Maximize 2x + 3y subject to x + y <= 10,
x, y >= 0."

LEVEL 2:  General Structure Linear Optimization Problem
- Features:  Described in natural language, requiring the translation
of the specific scenario into an objective function and constraints;
3-6 variables and constraints; constraints are direct resource
limitations or capacity conditions; no complex logical dependencies.
- Typical Problem Example:  "A factory produces three products with
limited raw materials, and the goal is to maximize profit."

LEVEL 3:  Optimization Problem with Implicit Logic and Conditional
Dependencies
- Features:  In addition to standard constraints, there are logical
dependencies/conditional restrictions (e.g., "at least two of the
following must be selected," "if A happens, B must not happen");
moderate number of variables/constraints (e.g., 5-10).
- Typical Problem Example:  "Given a limited advertising budget,
multiple channels must be chosen, but if TV is selected, social media
cannot be chosen, and at least three must be selected."

LEVEL 4:  Combinatorial Structure/Strong Logical Dependencies
Optimization Problem
- Features:  Multiple combinatorial constraints or strong logical
dependencies (e.g., facility location, employee scheduling); explicit
mutual exclusivity, inclusion, and dependency relations; difficult to
linearize directly from text.
- Typical Problem Example:  "A company is selecting warehouse
locations across several cities, each warehouse requires hiring
employees and incurring fixed costs, and must cover all customers
```

```
while the total budget does not exceed a limit."

LEVEL 5:  Large-Scale / Multi-Stage / Coupled Structure Optimization
Problem
- Features:  Multi-period, multi-stage, or multi-level nested
structures (e.g., dynamic inventory, supply chain scheduling, temporal
coupling); large number of variables and constraints (over dozens);
the objective may involve phase aggregation or balancing multiple
objectives.
- Typical Problem Example:  "Planning production and inventory for
the next six months, considering demand forecasts, inventory holding
costs, and production constraints, while smoothing monthly capacity
utilization." "
```

The detailed information of the dataset is shown in Table 5.

Table 5: Statistical information of the optimization datasets.

| Dataset | # of problems | Difficulty Count | | | | | Average Difficulty |
|---|---|---|---|---|---|---|---|
| | | 1 | 2 | 3 | 4 | 5 | |
| MamoEasy [10] | 100 | 5 | 93 | 2 | 0 | 0 | 1.97 |
| NL4Opt [19] | 100 | 2 | 80 | 18 | 0 | 0 | 2.16 |
| NLP4LP [2] | 100 | 1 | 71 | 23 | 5 | 0 | 2.32 |
| ComplexOR [23] | 19 | 0 | 13 | 0 | 4 | 1 | 2.61 |
| MamoComplex [10] | 100 | 0 | 37 | 25 | 25 | 13 | 3.14 |
| IndustryOR [22] | 100 | 1 | 31 | 27 | 25 | 16 | 3.24 |

- **MamoEasy** is the entry-level subset of the Mamo benchmark, comprising a collection of well-structured linear and mixed-integer linear programming (MILP) problems. Designed for basic training and algorithm validation, it provides foundational tasks for learning optimization modeling and solution strategies.

- **NL4Opt** is a natural language-based dataset that spans multiple real-world domains such as investment, advertising, and sales. It focuses on converting textual descriptions into optimization models, offering a valuable resource for research on text-to-model translation and automatic formulation generation.

- **NLP4LP** (Natural Language Processing for Linear Programming) collects classical LP problems from textbooks and lecture notes, covering topics like scheduling, network flow, and facility location. It integrates descriptive texts, structured data files, and reference solutions, supporting research in automated modeling and natural language understanding for optimization.

- **ComplexOR** is a curated dataset of high-complexity operations research (OR) problems sourced from domains such as logistics, scheduling, and supply chain management. It emphasizes diversity in problem types and real-world constraints, serving as a benchmark for evaluating algorithm robustness and modeling versatility.

- **MamoComplex** is the advanced-level counterpart to MamoEasy, featuring more challenging and theoretically rich MILP problems. It is tailored for upper-level coursework and research experiments, offering a broader spectrum of applications and higher-level optimization skills development.

- **IndustryOR** is the first dataset explicitly constructed for industrial optimization tasks. It draws from 13 industries and includes linear, integer, nonlinear, and special types of programming problems. Organized by difficulty levels, it provides a realistic and diverse benchmark for evaluating modeling and algorithmic generalization in industrial settings.

## A.2 Hyperparameters

Since our method is based on in-context learning with a frozen large language model, no parameter tuning or model training is involved. Instead, hyperparameter selection primarily concerns the configuration of the reasoning framework, particularly the parameters governing the Monte Carlo Tree Search (MCTS) process and prompt construction.

We adopted a heuristic-based approach to determine these hyperparameters. Initial values were informed by prior work on language model-based reasoning and were further refined through empirical evaluation on a development set. Key parameters such as the number of rollouts, the exploration-exploitation trade-off constant, prompt length, and temperature for response sampling were adjusted to balance reasoning depth, consistency, and computational efficiency. Hyperparameters were selected based on their ability to ensure stable decision paths, interpretable intermediate reasoning steps, and robust performance across diverse instances. To avoid overfitting to any single example, all tuning was done without access to test instances, and no gradients or updates were applied to the model. The final hyperparameter settings are shown in Table 6.

Table 6: Hyperparameter configuration.

| Hyperparameter Description | Value |
|---|---|
| Maximum number of components per expansion | 3 |
| Maximum number of nodes per layer | 5 |
| Maximum number of search iterations | 20 |
| Exploration weight of UCT $c$ | 2 |
| Reward weight $\alpha, \beta, \gamma$ | 0.1, 0.8, 0.1 |
| Local uncertainty threshold $\eta$ | 0.3 |
| LLM temperature | 0.2 |

# B  More Experimental Results

## B.1  Evaluation with a Lighter Language Model

While SolverLLM is originally implemented with GPT-4o as its underlying language model, we aim to examine the robustness and adaptability of the overall framework under reduced model capacity. In practical deployments, lighter language models may be preferred due to constraints on computational cost, latency, or API availability. Therefore, we evaluate the extent to which SolverLLM can retain its performance when powered by a smaller model, GPT-4o-mini [11]. We replicate the exact same experimental pipeline and evaluation protocol as in the main experiments, with the only difference being the replacement of the base language model. All other components of SolverLLM, including MCTS reasoning, uncertainty handling, and pruning strategies, remain unchanged. This ensures that any performance difference can be directly attributed to the change in language model capability, rather than to procedural or implementation differences.

As shown in Table 7, SolverLLM maintains strong performance across all datasets even when GPT-4o is replaced with the smaller GPT-4o-mini model. While a moderate performance drop is observed, the degradation remains limited, with all solving accuracies staying within a reasonable range. This indicates that the framework is not overly reliant on the specific language model scale and can generalize well under reduced model capacity.

These results underscore the robustness of the SolverLLM framework. Despite the lighter model's reduced language modeling capacity, the system maintains a high level of performance across diverse datasets. This resilience is attributable to the design of SolverLLM itself, which externalizes reasoning into an MCTS-driven planning process and relies on structural feedback (e.g., reward signals, execution validity) rather than blind reliance on model output fluency. In effect, the framework compensates for potential model weaknesses by grounding generation in semantically guided search. Consequently, SolverLLM exhibits graceful degradation under model compression and remains effective in resource-constrained environments.

Table 7: Comparison of SolverLLM with different LLM backends in terms of the SA metric.

| | MamoEasy | NL4Opt | NLP4LP | ComplexOR | MamoComplex | IndustryOR |
|---|---|---|---|---|---|---|
| SolverLLM with GPT-4o-mini | 94.0% | 94.0% | 81.0% | 72.2% | 66.0% | 48.0% |
| SolverLLM with GPT-4o | 96.0% | 97.0% | 87.0% | 77.8% | 76.0% | 56.0% |

## B.2 The Impact of Type Element on Graph-Based Problems

As part of the formulation process, the Type Element component is responsible for generating a global instruction that defines the overarching structure and semantics of the optimization problem. This global instruction serves as a high-level guide to constrain and contextualize the generation of all subsequent components in the formulation pipeline. While the Type Element benefits general task consistency, we observe that it is particularly impactful in the domain of graph-based problems, where problem structure is more abstract and constraints often depend on implicit topological relations.

Taking the Traveling Salesman Problem (TSP) as a representative graph problem, there are 25 TSP problems in the MamoComplex dataset. SolverLLM correctly solves 24 of them, achieving an accuracy of 96% on this problem type. In contrast, SolverLLM w/o TE (without the Type Element) solves only 16 instances correctly, resulting in a significantly lower accuracy of 64%. Below is an example of the global instructions automatically generated by SolverLLM:

---

**An example of global instructions for TSP**

```
1.  Define binary decision variables x[i, j] indicating whether the
path goes directly from city i to city j.
2.  Define integer variables u[i] for subtour elimination only for
non-starting cities (i.e., do not include the starting city in u).
3.  Choose one city as the starting point (e.g., 'A') and fix u['A'] =
0.
4.  Set bounds for u[i] as integers between 1 and (n - 1), where n is
the number of cities.
5.  Formulate the MTZ subtour elimination constraints as:  u[i] - u[j]
+ n * x[i, j] <= n - 1 for all i ≠ j, i ≠ start, j ≠ start
6.  Ensure that each city is visited exactly once (in-degree and
out-degree = 1).
7.  Use value() to print evaluated results instead of symbolic
expressions.
8.  Make sure the model does not include self-loops (i.e., no x[i, i])
and handles city sets and route sets automatically based on the input
list of cities.
```

---

These global descriptions play a crucial role in anchoring the meaning of variables, constraints, and objectives in downstream components. Without such context, the generation process may lack alignment with the structural properties of the underlying graph. For instance, in the absence of the Type Element, the TSP formulation often fails to recognize the need for subtour elimination and consequently omits the Miller-Tucker-Zemlin (MTZ) constraints, which are essential for preventing disconnected subcycles. Without this global instruction, the model lacks the structural understanding required to differentiate a valid tour from a set of disjoint paths, resulting in infeasible or incorrect solutions that violate the problem's core requirements.

This case study highlights the significance of the Type Element in structuring domain-specific inductive biases, particularly in tasks involving implicit topological or flow-related logic. It demonstrates that explicitly encoding problem type at the outset substantially improves semantic fidelity and solving accuracy, especially in structurally complex domains such as graph theory.

# C Model Implementation Details

## C.1 Implementation Details of Semantic Uncertainty Estimation

To quantify the semantic uncertainty associated with each candidate formulation node, we adopt an entropy-based measure grounded in recent work on meaning-equivalence in language model outputs. Rather than assessing variability over surface-level generations, we consider the distribution over underlying semantic classes—clusters of sequences that encode the same meaning. This approach captures uncertainty at the level of intended formulation semantics, aligning closely with the structural fidelity required in optimization tasks.

Formally, given a formulation $f_s$ and input context $x$, we sample a set of sequences $\{s\}$ from the language model and group them into equivalence classes $\{M\}$, where each class $M_i$ corresponds to a distinct semantic meaning. We then compute the *semantic entropy* $SE(x)$ [15], which serves as our measure of uncertainty, using the following definition:

$$SE(x) = -\sum_m p(m \mid x) \log p(m \mid x) = -\sum_m \left( \sum_{s \in m} p(s \mid x) \right) \log \left( \sum_{s \in m} p(s \mid x) \right).$$

This quantity reflects the dispersion of probability mass over meaning-equivalent outputs, with higher entropy indicating greater ambiguity or instability in the model's semantic preference.

Since the full distribution over all possible meaning classes is intractable, we approximate $SE(x)$ using Monte Carlo sampling over equivalence classes identified from sampled generations. Specifically, we estimate the expectation via:

$$SE(x) \approx -|M|^{-1} \sum_{i=1}^{|M|} \log p(M_i \mid x),$$

where $M_i$ denotes the $i$-th sampled equivalence class and $p(M_i \mid x)$ is its aggregated likelihood.

This entropy-based uncertainty measure serves as a principled signal during reward backpropagation in MCTS. Nodes associated with high semantic entropy are treated as less reliable and receive attenuated influence during search. Compared to naive variance-based metrics, semantic entropy offers a deeper view of model uncertainty grounded in meaning-level diversity, leading to improved robustness in structurally complex tasks.

## C.2 Feedback from Evaluation

LLM-generated feedback is a key component for SolverLLM. After evaluating a complete formulation, the feedback takes the following form and is subsequently used during the backpropagation phase:

**Feedback from evaluation**

```
{
 "score":  "a value between 0 and 100",
 "evaluation":  {
 "type":  {
 "need_revise":  "1 (True) or 0 (False)",
 "reason":  "Justification for the 'need_revise' decision",
 "prompt":  "Prompt that helps avoid the same problem in future",
 "uncertainty":  "Predictive entropy computed from the 'reason' field"
 },
 "sets":  "...  (same structure as above)",
 "parameters":  "...",
 "variables":  "...",
 "objective":  "...",
 "constraints":  "..."
 }
}
```

Among them, `score` is obtained by the following prompt:

---

**Partial prompt for objective score**

```
The task is to evaluate this formulation according to the following
criteria:
1.  Correctness:  Does the formulation correctly model the problem
described?
2.  Completeness:  Does the formulation include all necessary
components?
3.  Efficiency:  Is the formulation efficient in terms of variables
and constraints?
4.  Solvability:  Did the execution produce a correct solution?
5.  Solution quality:  Does the solution make sense for the given
problem?

Based on your evaluation, provide a numerical score from 0 to 100,
where:
- 0-20:  Poor formulation with major flaws
- 21-40:  Flawed formulation with significant issues
- 41-60:  Adequate formulation with some issues
- 61-80:  Good formulation with minor issues
- 81-100:  Excellent formulation that correctly models the problem
```

---

### C.3   Error Backpropagation for Code Generation

Inspired by LLMOPT [12], we incorporate an *error backpropagation* mechanism during the post-formulation code generation stage to enhance execution reliability. While SolverLLM focuses on producing high-quality optimization formulations via MCTS, code execution may still fail due to syntax issues, undefined references, or incompatibilities with solver requirements, even when the formulation itself is semantically correct.

To mitigate this, we implement a feedback-driven loop in the code generation process. Upon encountering a runtime error during execution, the corresponding error message is extracted and integrated into the next generation prompt as an explicit instruction. This instructive feedback helps the language model to revise the previous code in a targeted and informed manner. The process repeats until a valid, error-free program is generated or a predefined retry limit is reached. Following LLMOPT, we set the maximum number of retries to 12. An example of the prompt with error feedback for code generation is as follows.

---

**Prompt with error backpropagation for code regeneration**

```
The task is to implement the following mathematical formulation using
Pyomo.  The corresponding code has been generated by the large model
here, but it contains some errors.  Please generate accurate and error
free new code.

Problem description:
[problem_description]

Formulation:
[formulation_str]

The following Pyomo code was generated using a LLM:
[previous_pyomo_code]
```

---

```
There is an error in this code.  The error message is:
[error]

Please take into account the problem description, the mathematical
formulation, the previous Pyomo code, and the error message
comprehensively, and modify the previous Pyomo code accordingly to
generate a corrected version.

Please generate complete, executable Pyomo code that implements this
formulation.  The code should:
1.  Directly use "from pyomo.environ import *" as the import command.
2.  Create a concrete model
3.  Define sets, parameters, variables, objective, and constraints as
specified in the formulation
4.  Include code to solve the model with a solver.
5.  Display the results.
```

This mechanism yields multiple benefits. It significantly improves the Execution Rate (ER) by enabling recovery from execution failures through guided prompt revision. As a result, formulations that are structurally correct but fail during code translation are less likely to be discarded. In turn, this indirectly enhances the overall Solving Accuracy (SA) by preserving and executing more viable solutions. Furthermore, since the primary computational cost in SolverLLM arises during the MCTS-based formulation phase, ensuring successful execution helps avoid wasteful recomputation and improves overall resource efficiency.

## C.4 Pruning Module in the Dynamic Expansion Process

To ensure diversity and prevent redundancy, a pruning module removes semantically duplicate or overly similar nodes before expansion. This component plays a critical role in enhancing the quality and breadth of the search space during the formulation process. As each expansion step generates multiple candidate components via language model sampling, there is a high likelihood of producing repetitive or near-identical outputs due to the inherent sampling bias of large language models toward frequent or structurally similar patterns.

To address this, we introduce a pruning mechanism that operates immediately after each expansion step. The core idea is to evaluate pairwise similarity among candidate formulation components and discard those that are excessively similar to previously retained candidates. This filtering process ensures that only semantically distinct components are preserved, thereby promoting diversity and reducing redundancy in the search trajectory.

In practice, we adopt a string-level similarity metric to compare each newly generated component against those already selected. A candidate is accepted only if its similarity to all previously retained components remains below a predefined threshold. We set this threshold to 0.8, which empirically strikes a good balance between eliminating near-duplicates and retaining meaningful diversity across datasets and problem domains.

This pruning strategy offers several advantages. First, it encourages the exploration of varied and non-overlapping formulation structures, thereby improving the expressiveness and generality of the search space. Second, by filtering out redundant components early on, it improves the efficiency of downstream processes such as code generation and execution. Third, it helps avoid local saturation, where multiple candidate nodes compete to represent essentially the same solution fragment, leading to premature convergence.

## C.5 Hierarchical Prompting in the Search Process

To support structured reasoning over the search tree, we design a set of layer-specific prompts corresponding to the six levels of the search process. Each prompt is tailored to guide the generation of formulation components appropriate for its respective layer, while maintaining a consistent overall structure. In constructing these prompts, we prioritize generality over instance-specific

optimization, aiming to develop instructions that are broadly applicable across different tasks and domains. Furthermore, we simplify the prompts as much as possible to reduce cognitive and computational overhead, ensuring clarity without sacrificing effectiveness.

The prompt design for the the Type layer is as follows.

---

**Prompt for generation at the Type layer**

```
I need you to help me generate the TYPE component for a mathematical
optimization formulation.
You are also required to determine which category of classic
optimization problem the given instance belongs to.

Here's the problem description:
[problem_description]

Please judge the type of this optimization problem.  Return your
response as a Python list with a single dictionary.
* The dictionary should have a 'type' key, and its value must be one
of the following:  LP, MILP, or NLP.
* The dictionary should have a 'subtype' key, which indicates the
category of the classic problem this instance belongs to, such as a
production planning problem, a scheduling problem, or a traveling
salesman problem.
Don't include any explanations.

Here are some instructions for you:
Firstly, you need to determine whether the problem is linear.  The
rules are as follows:
* If the objective and constraints of the model involve non-linear
terms (such as power functions, multiplication, non-linear probability
models, etc.), then the problem is non-linear and returns directly to
NLP
* If the objective and constraint of the model are both linear, then
the problem is linear.  Furthermore, you need to determine whether the
problem is LP or MILP.

Here are the suggestions provided by experts based on the errors that
occurred during the previous generation of the TYPE component.  Please
take these suggestions into account during the generation process:
[suggestions]
```

---

The prompt design for the the Sets layer is as follows.

---

**Prompt for generation at the Sets layer**

```
I need you to help me generate the SETS component for a mathematical
optimization formulation.

Here's the problem description:
[problem_description]

Here is the type of problem that have already been defined:
[type_str]

Here are some instructions for solving this problem:
[instructions_str]
```

---

```
Please provide the sets needed for this optimization problem.  Return
your response as a Python list of dictionaries.
Each dictionary should have 'name', 'dimen' and 'elements' keys.
Don't include any explanations.

IMPORTANT: Do not use markdown formatting or code blocks.  Return only
the raw Python list.

Here are the suggestions provided by experts based on the errors that
occurred during the previous generation of the SETS component.  Please
take these suggestions into account during the generation process:
[suggestions]
```

The prompt design for the the Parameters layer is as follows.

**Prompt for generation at the Parameters layer**

```
I need you to help me generate the PARAMETERS component for a
mathematical optimization formulation.

Here's the problem description:
[problem_description]

Here is the type of problem that have already been defined:
[type_str]

Here are some instructions for solving this problem:
[instructions_str]

Here are the sets that have already been defined:
[sets_str]

Please provide the parameters needed for this optimization problem.
Return your response as a Python list of dictionaries.
For indexed parameters, include 'name', 'index_set', and 'values'
keys.
For scalar parameters, include 'name' and 'value' keys.
Don't include any explanations.

IMPORTANT: Do not use markdown formatting or code blocks.  Return only
the raw Python list.

Here are the suggestions provided by experts based on the errors that
occurred during the previous generation of the PARAMETERS component.
Please take these suggestions into account during the generation
process:
[suggestions]
```

The prompt design for the the Variables layer is as follows.

**Prompt for generation at the Variables layer**

```
I need you to help me generate the VARIABLES component for a
mathematical optimization formulation.

Here's the problem description:
[problem_description]
```

```
Here is the type of problem that have already been defined:
[type_str]

Here are some instructions for solving this problem:
[instructions_str]

Here are the sets that have already been defined:
[sets_str]

Here are the parameters that have already been defined:
[parameters_str]

Please provide the variables needed for this optimization problem.
Return your response as a Python list of dictionaries.
Each dictionary should have 'name', 'domain', and optionally
'index_set' and 'description' keys.
The domain should be one of:  'Binary', 'Integer',
'NonNegativeIntegers', 'NonNegativeReals', or 'Reals'.
Don't include any explanations.

IMPORTANT: Do not use markdown formatting or code blocks.  Return only
the raw Python list.

Here are the suggestions provided by experts based on the errors that
occurred during the previous generation of the VARIABLES component.
Please take these suggestions into account during the generation
process:
[suggestions]
```

The prompt design for the the Objective layer is as follows.

**Prompt for generation at the Objective layer**

```
I need you to help me generate the OBJECTIVE component for a
mathematical optimization formulation.

Here's the problem description:
[problem_description]

Here is the type of problem that have already been defined:
[type_str]

Here are some instructions for solving this problem:
[instructions_str]

Here are the sets that have already been defined:
[sets_str]

Here are the parameters that have already been defined:
[parameters_str]

Here are the variables that have already been defined:
[variables_str]

Please provide the objective function for this optimization problem.
Return your response as a Python list with a single dictionary.
The dictionary should have 'name', 'sense', and 'expression' keys.
The sense should be either 'maximize' or 'minimize'.
```

The expression should be a valid Pyomo expression as a string, using
'model.' to reference sets, parameters, and variables.
Don't include any explanations.

IMPORTANT: Do not use markdown formatting or code blocks. Return only
the raw Python list.

Here are the suggestions provided by experts based on the errors that
occurred during the previous generation of the OBJECTIVE component.
Please take these suggestions into account during the generation
process:
[suggestions]

The prompt design for the the Constraints layer is as follows.

**Prompt for generation at the Constraints layer**

I need you to help me generate the CONSTRAINTS component for a
mathematical optimization formulation.

Here's the problem description:
[problem_description]

Here is the type of problem that have already been defined:
[type_str]

Here are some instructions for solving this problem:
[instructions_str]

Here are the sets that have already been defined:
[sets_str]

Here are the parameters that have already been defined:
[parameters_str]

Here are the variables that have already been defined:
[variables_str]

Here is the objective function:
[objective_str]

Please provide the constraints for this optimization problem.  Return
your response as a Python list of dictionaries.
Each dictionary should have 'name', 'expression', and optionally
'description' keys.
The expression should be a valid Pyomo expression as a string, using
'model.' to reference sets, parameters, and variables.
Don't include any explanations.

IMPORTANT: Do not use markdown formatting or code blocks.  Return only
the raw Python list.

Here are the suggestions provided by experts based on the errors that
occurred during the previous generation of the CONSTRAINTS component.
Please take these suggestions into account during the generation
process:
[suggestions]

## D  Limitations

While SolverLLM achieves strong performance across diverse optimization tasks without any training or fine-tuning, there remain several limitations. First, the current framework may incur relatively high inference latency due to its reliance on Monte Carlo Tree Search (MCTS), especially on complex problems with large formulation spaces. While this cost is offset by its generalization ability and test-time adaptability, it may limit real-time applications. Second, while the framework incorporates uncertainty-aware mechanisms to address the inherent noisiness of LLM outputs, the effectiveness of reward estimation remains bounded by the subjective and non-deterministic nature of language models. Third, the current evaluation is restricted to datasets with well-structured and moderately constrained optimization problems. The robustness of SolverLLM under highly ambiguous, adversarial, or noisy natural language descriptions remains an open area for exploration. These limitations, while not critical to the core contributions, suggest several promising directions for future research.

## E  Broader Impact

SolverLLM proposes a novel inference-time framework that combines LLM reasoning with algorithmic search to solve optimization problems without supervised training. This paradigm has the potential to make optimization modeling more accessible to non-experts and reduce the dependency on curated datasets. It could benefit a wide range of fields, including operations research, supply chain management, and education, by enabling users to express complex optimization needs in natural language. However, the use of LLMs for automated decision modeling also raises concerns. In high-stakes applications such as healthcare or infrastructure planning, even small errors in formulation can propagate through automated pipelines and lead to unintended outcomes. Furthermore, reliance on opaque LLM outputs for critical tasks may pose challenges for interpretability and accountability. As such, we advocate for cautious deployment, with appropriate mechanisms for human-in-the-loop verification and domain-specific validation.

