# OpenReview forum: "SolverLLM: Leveraging Test-Time Scaling for Optimization Problem via LLM-Guided Search"
_NeurIPS.cc/2025/Conference — NeurIPS 2025 poster_

### Official Review · Reviewer_6kR9 · 2025-06-25

**Clarity:** 2
**Significance:** 2
**Originality:** 2
**Rating:** 3
**Confidence:** 3

**Summary:**

SolverLLM is a training-free, test-time–scaling framework that pairs a large language model with Monte Carlo Tree Search to translate natural-language optimization problems into mathematical formulations and solver-ready code.  Its search adds *dynamic expansion* (revisiting and refining partial formulations), *prompt backpropagation* (using solver feedback to guide later LLM prompts), and *uncertainty backpropagation* (down-weighting noisy rewards), all organized around a six-element schema (Type, Sets, Parameters, Variables, Objective, Constraints).  Across six standard benchmarks, SolverLLM surpasses state-of-the-art prompt- and learning-based baselines—improving solving accuracy by up to 10 percentage points while preserving near-100 % code-execution rates, and it does so without any additional training data or fine-tuning.

**Questions:**

* Section 3.2.1 still leaves the relationship between nodes granularity, the six-element schema, and a “complete path” ambiguous. Does one node correspond to a single element (e.g., Variables) or to a full partial formulation spanning several elements? What exactly constitutes a terminal path in the tree?
* What is x^* on line 174? explain more about the objective_score and "Optimality" factor. Is there a solver output as ground truth ? what does it mean by "estimate relative solution quality even in the absence of exact ground-truth labels or reference objectives"?
* How were the MCTS expansion depth, rollout count, and UCB hyper-parameters chosen? A sensitivity study would show whether SolverLLM’s gains persist under tighter time or compute budgets and ensure a fair comparison to prompt-only baselines.
* How are all uncertainty signals combined, and why are all three necessary? Could they conflict or over-count uncertainty? Empirical evidence isolating each contribution would help establish whether the added complexity is warranted.

**Ethical Concerns:**

["NO or VERY MINOR ethics concerns only"]

**Final Justification:**

The rebuttal addressed some of my technical concerns, clarifying resource trade-offs, uncertainty mechanisms, and methodological details with helpful ablations. While I remain cautious about the overall complexity, limited evaluation breadth, and incremental novelty, the responses improved my confidence. I am therefore raising my score, though with some reservations about broader impact. However, due to some personal matters, I was unable to engage in deeper discussion during the rebuttal period, so I am lowering my confidence score.

**Limitations:**

yes

**Quality:**

3

**Strengths And Weaknesses:**

# Strengths

* Achieves up to 10 pp absolute accuracy gains over state-of-the-art prompt- and learning-based baselines while preserving near-100 % executable code, showing practical value for real-world optimization tasks.
* Integrates Monte Carlo Tree Search with three novel feedback mechanisms (dynamic expansion, prompt & uncertainty back-propagation) that demonstrably boost solving accuracy without any extra training.

# Weaknesses

* MCTS plus iterative solver calls increase test-time latency and resource consumption. The paper would benefit from concrete runtime and cost statistics to contextualize this overhead
* I do understand that detail steps of MCTS could be complicate. Despite the detailed write-up already, several steps of the MCTS procedure remain opaque and would profit from clearer explanations or illustrative examples (see questions).
* The design of prompt back-propagation offers limited novelty, as similar ideas—such as prompt reflection—have been explored in prior work [1][2].
* The design of uncertainty back-propagation is somewhat confusing. First, the method already employs the UCB algorithm to balance exploration and exploitation during the selection phase. Second, it uses predictive entropy during prompt back-propagation to determine whether a node should be “activated” for dynamic expansion; however, predictive entropy is generally not a reliable uncertainty metric for guiding selection [3] and it entangles with the UCB equation. Third, it introduces yet another uncertainty estimate by computing a semantic uncertainty score based on the variance of judge rewards from the LLM, and then applies a heuristic reweighting scheme using this variance. The interaction among these three overlapping uncertainty mechanisms feels unnecessarily complex and poorly motivated. Moreover, the repeated sampling required by this setup leads to significant computational overhead. According to Table 3, the ablation study suggests that uncertainty back-propagation contributes little to the overall performance.
* The experimental comparison of prompt-based methods would be stronger if it incorporated additional MCTS-based baselines, which are more directly relevant to the proposed approach.



1. Shinn, Noah, et al. "Reflexion: Language agents with verbal reinforcement learning." Advances in Neural Information Processing Systems 36 (2023): 8634-8652.
2. Zhou, Andy, et al. "Language agent tree search unifies reasoning acting and planning in language models." arXiv preprint arXiv:2310.04406 (2023).
3. Guo, Pei-Fu, Yun-Da Tsai, and Shou-De Lin. "Benchmarking Large Language Model Uncertainty for Prompt Optimization." arXiv preprint arXiv:2409.10044 (2024).

---

> ### Author Rebuttal · Authors · 2025-07-31
>
> We sincerely thank Reviewer 6kR9 for the thoughtful feedback and for recognizing both the practical value of our work in real-world scenarios and the methodological innovations. The concerns raised mainly stem from questions about methodological complexity, additional overhead, and the details and motivations of certain mechanisms. We provide point-by-point clarifications below.
>
> > ### [W1] Concern regarding the resource consumption.
>
> The core idea of test-time scaling methods is to trade increased computation at test/inference time (rather than more training data or larger model parameters) for better performance. This inevitably increases test-time latency and resource consumption. Our paper already includes analysis of Average Generation Times (AGT) at 20 search iterations. Additionally, we provide statistics on token budget, Solving Accuracy (SA), and AGT of SolverLLM under different numbers of search iterations on the MamoComplex dataset:
>
> | # Search Iterations | Token Budget | SA (%) | AGT (min) |
> |-------------------|--------------|--------|-----------|
> | 10                | 32,790       | 69.0   | 2.54      |
> | 20                | 40,920       | 76.0   | 3.85      |
> | 30                | 49,337       | 77.0   | 4.82      |
> | 40                | 57,834       | 78.0   | 5.98      |
> | 50                | 66,312       | 78.0   | 6.73      |
>
> It can be observed that there is indeed a trade-off between cost and performance. After balancing these factors, we chose 20 as the number of search iterations to achieve an acceptable cost consumption.
>
> > ### [W2] Concern about the detailed methodology.
>
> Please refer to the answer to the questions.
>
> > ### [W3] Concern about the novelty of prompt backpropagation.
>
> We thank the reviewer for highlighting the relevance to Reflexion [1] and LATS [2]. While Prompt Backpropagation (PB) shares the idea of using feedback to refine reasoning, its core innovation is enabling **Dynamic Expansion**, allowing the generation of new formulation candidates in an open-ended and context-aware manner. Our approach differs significantly in objectives and mechanisms scope:
>
> 1. **Objectives**
>
> + Reflexion [1] focuses on language-agent tasks via verbal reinforcement.
> + LATS [2] unifies acting and reasoning in planning through tree search.
> + Our work addresses solver-ready formulation construction for optimization, requiring both constraint feasibility and optimization objectives—challenges beyond standard language reasoning.
>
> 2. **Mechanisms**
>
> + PB integrates structured evaluation with uncertainty modeling, maintaining a node-level knowledge base to guide future generations.
> + Unlike Reflexion’s single verbal feedback, PB forms a training-free reasoning mechanism deeply coupled with MCTS.
> + Compared to LATS, PB provides MCTS with the critical Dynamic Expansion capability, innovating across the entire search process.
>
> In summary, PB is not merely an adaptation of prompt reflection but a solver-aware, uncertainty-driven framework purpose-built for training-free optimization modeling.
>
>
> > ### [W4 & Q4] Concern about the introduction of uncertainty.
>
> We use three types of uncertainty, each serving a distinct role without conflict:
>
> 1. Uncertainty in Original UCB
>
> The original UCB’s exploration term increases uncertainty for less-visited nodes, encouraging exploration of under-explored branches.
>
> 2. Global Uncertainty
>
> LLM outputs, such as objective scores, have high variance causing unstable reward propagation. We use a sampling-based method to estimate semantic uncertainty as global uncertainty, which adjusts the UCB’s exploitation term via an uncertainty-weighted average, stabilizing backpropagation with controllable overhead.
>
> 3. Local Uncertainty
>
> During Prompt Backpropagation, we estimate sentence-level uncertainty from LLM feedback using predictive entropy to assess reasoning reliability at the layer level. This local uncertainty helps filter out unreliable instructions during search, but it does not affect UCB calculations or node rewards.
>
> Overall, the uncertainties in the two UCB terms are computed independently, while local uncertainty exclusively supports the Prompt Backpropagation process.
>
> The ablation study of the w/o UB variant shows that global uncertainty helps reduce time cost and slightly improve performance on more complex datasets, which is an important feature for real-world optimization. For local uncertainty, we additionally conducted experiments with a w/o local uncertainty variant on the MamoComplex dataset:
>
> | Method                | SA (%) | ER (%) | AGT (min) |
> |------------------------|--------|--------|-----------|
> | SolverLLM             | 76.0   | 99.0   | 3.85      |
> | w/o local uncertainty | 74.0   | 99.0   | 4.08      |
>
> The results indicate that local uncertainty reduces the generation of uncertain or incorrect instructions, thereby improving solving accuracy while lowering time consumption. We will include the complete results of this ablation in the final version.
>
>
> > ### [W5] Concern regarding the MCTS-based baselines.
>
> AutoFormulation serves as the MCTS-based baseline, against which we compared SolverLLM and achieved superior performance. To the best of our knowledge, up to the completion of this work, no new MCTS-based test-time scaling methods for optimization problem have been proposed.
>
> > ### [Q1] Question about the relationship between nodes granularity, the six-element schema, and a "complete path".
>
> The relationship between nodes, elements, and complete paths is best understood in the context of the full Monte Carlo Search Tree, as illustrated in Figures 4 and 5 of our paper. The search tree consists of six levels beyond the root, corresponding to the **six elements**: Type, Sets, Parameters, Variables, Objective, and Constraints. **Each node** corresponds to a single element, representing one possible modeling choice for that element conditioned on all its preceding nodes. Its main attributes include cumulative reward, visit count, partial formulation, and local uncertainty (see the `FormulationNode` class in `MCTS.py` for details). A partial formulation captures only the modeling of the current element; for instance, a node in the *Variables* level includes the variable formulation built upon the partial path formed by the preceding *Type, Sets,* and *Parameters* nodes. In contrast, a **complete path** refers to the sequence from the root to a leaf in the *Constraints* layer, encompassing one node from each level, thereby constructing a full solver-ready formulation of the optimization problem under the six-element schema.
>
> > ### [Q2] Question about the reward in Simulation stage.
>
> Here, $x^{\*}$ denotes the output obtained by executing the formulation $f_s$, which has been translated into code and solved by a numerical solver. The **objective\_score** function can be understood as prompting the LLM to act as a lightweight evaluator (or “judger”), leveraging both $f_s$ and $x^{\*}$ to assign a scalar score between 0 and 100. The goal is to assess the *optimality* of $f_s$, that is, how well the solution aligns with the intent and structure of the formulation. A portion of the prompt is as follows:
>
> ```
> The task is to evaluate this formulation according to the following criteria:
> 1. Correctness: Does the formulation correctly model the problem described?
> 2. Completeness: Does the formulation include all necessary components?
> 3. Efficiency: Is the formulation efficient in terms of variables and constraints?
> 4. Solvability: Did the execution produce a correct solution?
> 5. Solution quality: Does the solution make sense for the given problem?
>
> Based on your evaluation, provide a numerical score from 0 to 100, where:
> - 0-20: Poor formulation with major flaws
> - 21-40: Flawed formulation with significant issues
> - 41-60: Adequate formulation with some issues
> - 61-80: Good formulation with minor issues
> - 81-100: Excellent formulation that correctly models the problem
> ```
>
> The reason we describe this as *estimating relative solution quality even in the absence of exact ground-truth labels or reference objectives* is that the true ground-truth for each problem is unknown. Nevertheless, after each simulation generates a complete formulation, an evaluation is still necessary to assess its quality. Thus, we employ relative quality assessment based on the five dimensions of **Correctness, Completeness, Efficiency, Solvability, and Solution Quality**.
>
> We will further refine this explanation in the final version to make the rationale more explicit, and we sincerely thank the reviewer for pointing this out.
>
> > ### [Q3] Question about the hyper-parameters.
>
> The expansion depth of MCTS depends on the formulation modeling approach. As mentioned in Q1, we model the search tree as a six-level structure excluding the root node. Regarding the rollout count and UCB hyperparameters, we use a heuristic-based approach to determine them, with specific details provided in Appendix A.2 for reference.
>
> We conducted a sensitivity analysis of the exploration weight $c$ in the UCB formula on the MamoComplex dataset, with the following results:
>
> | c   | SA (%) | AGT (min) |
> |-----|--------|-----------|
> | 0.1 | 69.0   | 2.75      |
> | 0.5 | 73.0   | 3.21      |
> | 1   | 74.0   | 3.68      |
> | 2   | 76.0   | 3.85      |
> | 5   | 77.0   | 4.45      |
> | 10  | 79.0   | 5.67      |
>
> We observe that larger values of $c$ encourage more exploration, i.e., modeling new formulations. This improves performance but increases time consumption. Smaller $c$ favors reusing well-performing formulations, reducing time but limiting exploration of harder problems. Balancing this trade-off, we chose $c=2$ as UCB’s exploration weight. This analysis will be included in the final version.
>
> We sincerely thank Reviewer 6kR9 for the valuable comments and hope our responses address your concerns. We look forward to further discussions to clarify any outstanding issues.

---

> ### Author Response · Authors · 2025-08-08
>
> We sincerely thank you for the time and effort you have dedicated to reviewing our paper. We kindly follow up to see if you might have any further thoughts or clarifications regarding our rebuttal. Your additional feedback would be greatly appreciated.

---

### Official Review · Reviewer_2ytH · 2025-06-30

**Clarity:** 3
**Significance:** 2
**Originality:** 3
**Rating:** 4
**Confidence:** 4

**Summary:**

This paper proposes a test-time computation scaling method for solving optimization problem with LLMs. The method is built upon MCTS and has three key improvement. First, the authors extend the five-elements abstraction to include another dimension of problem class, which allow the LLM to focus on particular problem types in subsequent modeling. Second, the proposed method allows an internal node to be eligible for expansion if it is marked as active node by the critic. Finally, the authors accumulate the textual feedbacks from critic to guide the further revisions for each layer. Besides, there are also some minor improvements like taking uncertainty into the considerations. Experimental results confirm the merits of the proposed methods.

**Questions:**

See my comments on Strengths And Weaknesses section.

**Ethical Concerns:**

["NO or VERY MINOR ethics concerns only"]

**Final Justification:**

The authors' rebuttal addressed my concerns. I therefore maintained my score.

**Limitations:**

yes

**Paper Formatting Concerns:**

N.A.

**Quality:**

2

**Strengths And Weaknesses:**

## Strength
- Generally, the paper is well-written. The contribution is clearly presented (except for prompt backpropagation point)
- The inclusion of type element is sensible, serving as a general guideline to narrow down the possible modeling choices for LLMs
- The use of active nodes allows dynamically retrying at the specific levels, which is also a sensible design choice
- The integration of knowledge-base for each level provides insights for revising incorrect formulations, akin to improvement-based local search

## Weakness
- The significance and novelty seem limited to me. Although all improvements to vanilla MCTS seem sensible, they are considered to be moderately important and novel from my perspective.
- Prompt backpropagation is somewhat misleading. If I understand correctly, the authors use the critic to judge each node in the currently selected reasoning path, and then store the textual feedback to the knowledgebase of each level. If it is the case, there is no "backpropagation" happened because the operation can be localized into each individual layer, as long as the partial formulation and the full consequence visible to the critic.
- The experiments are also not fully clear. How do you terminate your MCTS? How uncertainty threshold $\eta$ affects the performance? Also, I am expecting the results of token consumption of each methods, which is crucial to ensure a fair comparison for test-time scaling methods.
- A minor note: In line 216, $\lambda_s$ should be $\gamma_s$.

---

> ### Author Rebuttal · Authors · 2025-07-30
>
> We sincerely thank Reviewer 2ytH for the careful reading of our paper and for recognizing the overall quality of writing, as well as the soundness, practicality, and ingenuity of our method. The concerns raised mainly stem from questions about the significance of our work and specific methodological details, which we clarify in the following response.
>
> > ### [W1] Concern regarding the significance and novelty.
>
> We understand the reviewer’s concerns regarding the novelty of our method and would like to clarify the challenges and contributions of our work.
>
> 1. Uniqueness of the research challenge
>
> Automatically converting natural language descriptions into high-quality optimization formulations is an inherently complex and long-standing open problem. It requires not only accurate semantic understanding but also guarantees that the generated formulations are executable by solvers and yield high-quality solutions. Compared to directly generating solutions, this task poses greater challenges in structured expression, cross-task generalization, and error correction.
>
> 2. Novelty of our method
>
> Our contribution lies not only in improvements to MCTS itself but, for the first time, in combining dynamic expansion, prompt backpropagation, and uncertainty propagation into a test-time reasoning framework that synergizes with LLMs.
>
> + Dynamic Expansion overcomes the traditional MCTS limitation of expanding only at leaf nodes, allowing formulation corrections at any tree leve and a larger search space to help solve more complex problems;
> + Prompt Backpropagation transforms solver feedback into reusable semantic instructions that directly guide subsequent generation, a mechanism unexplored in existing MCTS-LLM integrations;
> + Uncertainty Propagation explicitly models LLM output confidence during search, significantly enhancing inference stability and efficiency.
>
> 3. From vanilla MCTS to a training-free framework
>
> Rather than making incremental improvements to vanilla MCTS, we extend it into an unsupervised, training-free framework that deeply couples with LLMs to tackle cross-domain optimization modeling without reliance on additional annotation or fine-tuning. This approach offers a novel paradigm for applying test-time scaling methods to complex decision-making problems.
>
> We believe overcoming these challenges and designing such a framework is not only technically innovative but also opens new possibilities for automation and generalization in optimization modeling.
>
> > ### [W2] Concern about prompt backpropagation.
>
> SolverLLM does not evaluate each node individually; instead, the evaluation of a formulation is based on the complete formulation. A partial formulation at a single level is meaningless on its own, which explains why the Simulation step is necessary in each MCTS iteration. Only with a complete formulation can we obtain a final score and evaluation used to assess the nodes along the path that constructed the current formulation.
>
> We integrate Prompt Backpropagation into the Backpropagation phase of MCTS iterations. The critic can judge each node on the currently selected reasoning path only after a complete reasoning path has been obtained. Subsequently, the evaluation results, which consist of textual instructions, rewards, and uncertainty, are backpropagated accordingly.
>
> > ### [W3] Concern about the experiments.
>
> We appreciate the reviewer’s thoughtful questions and will address them one by one:
> + [Q: How do you terminate your MCTS? ] SolverLLM terminates either when reaching the maximum number of search iterations or when a full $n$-ary tree with the maximum number of nodes per layer $n$ is generated.
> + [Q: How uncertainty threshold $\eta$ affects the performance?] We supplemented a sensitivity analysis of the uncertainty threshold $\eta$ on the MamoComplex and IndustryOR datasets, as shown below:
>
> **MamoComplex:**
> | η    | SA (%) | AGT (min) |
> |-----|---------|-----------|
> | 0.1 | 79.0    | 4.58      |
> | 0.2 | 78.0    | 4.22      |
> | 0.3 | 76.0    | 3.85      |
> | 0.4 | 73.0    | 3.54      |
> | 0.5 | 72.0    | 3.23      |
>
> **IndustryOR:**
> | η   | SA (%) | AGT (min) |
> |-----|---------|-----------|
> | 0.1 | 57.0    | 4.21      |
> | 0.2 | 56.0    | 3.54      |
> | 0.3 | 56.0    | 3.28      |
> | 0.4 | 54.0    | 3.12      |
> | 0.5 | 53.0    | 2.96      |
>
> It can be observed that the uncertainty threshold $\eta$ acts as a trigger: a lower $\eta$ produces more local instructions, increasing computational cost but improving performance, while a higher $\eta$ has the opposite effect.
>
> + [Q: What's the results of token consumption of each test-time scaling methods?] We compared the token consumption of SolverLLM and AutoFormulation, two test-time scaling methods, as shown below:
>
> | # Search Iterations | SolverLLM (# Tokens / SA %) | AutoFormulation (# Tokens / SA %) |
> |-------------------|---------------------------|---------------------------------|
> | 10                | 32,790 / 69.0            | 35,911 / 34.0                  |
> | 20                | 40,920 / 76.0            | 43,150 / 37.0                  |
> | 30                | 49,337 / 77.0            | 54,427 / 38.0                  |
> | 40                | 57,834 / 78.0            | 62,248 / 38.0                  |
> | 50                | 66,312 / 78.0            | 70,245 / 40.0                  |
>
> SolverLLM achieves better performance than AutoFormulation while using fewer tokens at the same number of search iterations due to a more efficient search strategy. In the final version, we will explicitly report the token consumption curves over varying search budgets to further enhance result transparency and comparability.
>
> > ### [W4] Concern regarding a typo.
>
> Thank you for your correction. We will address this issue in the final version.
>
> We sincerely thank the reviewer 2ytH for the thoughtful feedback and look forward to further addressing any remaining questions and engaging in constructive discussion.

---

> > ### Comment · Reviewer_2ytH · 2025-08-05
> >
> > Thanks for your responses which address my concerns

---

> > > ### Author Response · Authors · 2025-08-08
> > >
> > > Thank you for acknowledging that our responses have addressed your concerns. We greatly appreciate your thoughtful evaluation and constructive comments. Your feedback has been instrumental in helping us refine our work, and we remain committed to further improving the system based on the insightful points raised during the discussion.

---

### Official Review · Reviewer_SNpT · 2025-07-03

**Clarity:** 3
**Significance:** 1
**Originality:** 3
**Rating:** 4
**Confidence:** 4

**Summary:**

SolverLLM couples LLMs with an MCTS-style search to auto-generate mathematical programs from Natural Language descriptions, then hands the resulting model to Gurobi/Pyomo solvers for solution. Experiments on six academic benchmarks show higher solving accuracy than prior prompt-engineering baselines.

**Questions:**

1. Can you show Industry OR results with problems cluster in easy-hard? It would be better to understand where the gain comes from in real-world problems.

2. Scalability remains speculative.

2. What guardrails prevent LLM hallucinations from generating infeasible or numerically unstable formulations that crash the solver?

**Ethical Concerns:**

["NO or VERY MINOR ethics concerns only"]

**Final Justification:**

Please refer to the reply to authors.

**Limitations:**

Please refer to questions.

**Quality:**

3

**Strengths And Weaknesses:**

### Strengths

* Integrates three search refinements (dynamic expansion, prompt back-propagation, uncertainty weighting) into a coherent framework and ablates each component.

* Demonstrates consistent accuracy gains over existing prompt-based methods on benchmark datasets.

### Weaknesses

* **No solver-algorithm contribution.** Because the method defers all heavy lifting to a commercial solver, it adds nothing to algorithmic advances. If the solver already supports the model class, SolverLLM merely adds minutes of latency;

* **No comparison on solver time cost**. There is no evidence that the solverLLM could provide problem formulation which requires much less time to solve than human experts.

---

> ### Author Rebuttal · Authors · 2025-07-30
>
> We sincerely thank Reviewer SNpT for recognizing the completeness and systematic design of our approach, as well as the validity and consistency of our experimental results. The concerns raised mainly stem from potential challenges in deploying our work to real-world applications. We provide clarifications on these points below.
>
> > ### [W1] Concern about solver-algorithm contribution.
>
> We appreciate the reviewer’s concerns and would like to clarify that the primary contribution of our work does not lie in improving the underlying numerical solvers, but rather in enabling the automatic construction of high-quality, solver-ready formulations from natural language descriptions in a training-free manner. This has long been a bottleneck in applying optimization to real-world tasks. Our contributions can be summarized as follows:
>
> 1. Independence from specific problem domains
> While commercial solvers indeed excel in numerical optimization, they cannot directly handle problems specified in natural language. The core contribution of SolverLLM is a novel LLM-guided MCTS reasoning framework that dynamically generates, revises, and verifies formulations at inference time, thereby enabling automated and cross-domain optimization.
>
> 2. Methodological novelty
> Unlike simple prompt engineering or supervised fine-tuning, SolverLLM introduces several inference-level innovations:
>
> + Dynamic Expansion: allowing revisions at non-leaf nodes during the search process;
> + Prompt Backpropagation: leveraging solver feedback to guide subsequent prompt construction;
> + Uncertainty Backpropagation: explicitly modeling the confidence of LLM feedback to enhance search efficiency.
> These mechanisms constitute substantive algorithmic contributions in reasoning and search, beyond straightforward interface calls.
>
> 3. On additional latency
> The essence of test-time scaling is to trade off inference-time cost for reduced training-time human effort, thereby achieving training-free deployment. Such methods have been widely adopted and accepted in recent LLM research [1]. Our experiments (see Table 3) show that, thanks to uncertainty-guided pruning, the average additional latency remains within a few minutes, while achieving over a 10% accuracy improvement on complex datasets compared to traditional baselines. This demonstrates that the latency introduced by test-time scaling is both reasonable and practically acceptable.
>
> In summary, the value of SolverLLM lies in introducing a new test-time reasoning framework that enables high-quality modeling and solving of optimization problems without requiring training or additional labeled data. This is complementary to advances in numerical solvers rather than a mere additive improvement.
>
> > ### [W2] Concern about comparison with human experts.
>
> SolverLLM is designed as a tool accessible to non-experts while minimizing both training and human costs. Our focus is primarily on the algorithmic level, and we have compared our method with a wide range of existing approaches (including GPT-4 and GPT-4o, prompt-based methods such as Reflexion, Chain-of-Experts, and OptiMUS, as well as learning-based methods like ORLM and LLMOPT, and test-time scaling methods AutoFormulation) to demonstrate its superiority. Notably, these methods, including AutoFormulation which is directly comparable as a test-time scaling approach, also do not report comparisons against human experts.
>
> > ### [Q1] Can you show Industry OR results with problems cluster in easy-hard? It would be better to understand where the gain comes from in real-world problems.
>
> Based on the difficulty levels defined in Appendix A.1, we report SolverLLM’s Solving Accuracy (SA) on the IndustryOR dataset under different difficulty levels, as shown below (a higher value indicates a more difficult problem):
>
> | Difficulty Level | Amount | # Correct | SA (%)  |
> |------------------|--------|---------|---------|
> | 1                | 1      | 1       | 100.00 |
> | 2                | 31     | 25      | 80.64  |
> | 3                | 27     | 18      | 66.67  |
> | 4                | 25     | 8       | 32.00  |
> | 5                | 16     | 4       | 25.00  |
>
> As shown, SolverLLM achieves high accuracy on easier problems (Level 1-2) and maintains relatively high accuracy at moderate difficulty levels (Levels 3). While the accuracy decreases for the most challenging problems (Levels 4–5), SolverLLM still demonstrates the capability to provide feasible solutions, indicating its robustness across a wide range of problem complexities.
>
> > ### [Q2] Scalability remains speculative.
>
> Our method exhibits strong scalability.
>
> 1. Methodological perspective: The MCTS framework inherently supports scalability due to its anytime nature, allowing solution quality to improve with additional computational budget. By directing exploration toward high-value branches via UCT instead of performing exhaustive search, it efficiently manages large problem spaces. Its structure naturally supports parallelization and dynamic expansion, ensuring that only necessary nodes are explored, making it particularly suitable for real-world, large-scale problems.
>
> 2. Empirical perspective: We further validated scalability by rigorously evaluating our method on every problem across all datasets. Specifically, each problem was run three independent times, and a problem was considered solved only when all three runs produced a feasible and correct solution. This repeated and consistent success across diverse instances demonstrates that our approach is not only effective in isolated cases but also robust and scalable, ensuring reliable performance as problem complexity increases.
>
> > ### [Q3] What guardrails prevent LLM hallucinations from generating infeasible or numerically unstable formulations that crash the solver?
>
> We address this concern through multiple guardrails to ensure execution reliability. Specifically, we employ an error propagation mechanism (detailed in Appendix C.2) that detects and propagates solver execution errors back to the LLM, enabling corrective reformulation and making the approach naturally extendable to infeasible cases. To mitigate numerically unstable formulations, we impose prompt-level constraints on the LLM’s outputs and perform strict post-processing checks to enforce the required format and validity. Together, these mechanisms substantially reduce the risk of hallucinations leading to infeasible or unstable formulations that could otherwise crash the solver.
>
> We thank the reviewer SNpT for the valuable insights and look forward to clarifying any outstanding issues through further discussion.
>
> [1] Zhang Q, Lyu F, Sun Z, et al. A Survey on Test-Time Scaling in Large Language Models: What, How, Where, and How Well? arXiv preprint arXiv:2503.24235, 2025.

---

> > ### Comment · Reviewer_SNpT · 2025-08-05
> >
> > Thanks for the authors' detailed response.
> >
> > Most of my concerns are addressed, especially the significance of training LLM for optimization problem formulation, which could help daily LLM users to solve optimization problems by function calling.
> >
> > However, I still have concern in the performance on hard industrial examples, the MCTS back propagate does not seem to be effective on these scenarios.
> >
> > Therefore, I raise my score to 4.

---

> > > ### Author Response · Authors · 2025-08-08
> > >
> > > Thank you for your positive feedback and for raising your score. We are pleased that our response addressed most of your concerns, particularly regarding the significance of training LLMs for optimization problem formulation via function calling, which we believe is crucial for enhancing usability in real-world scenarios.
> > >
> > > Regarding your remaining concern on SolverLLM’s performance on hard industrial examples, we appreciate your continued attention to this aspect. In the current version, we have conducted ablation studies and provided detailed case analyses on challenging datasets to demonstrate the model's effectiveness under complex conditions. These results serve as preliminary evidence of SolverLLM’s potential in industrial-grade applications. Additionally, we are actively exploring more robust strategies, including better tree search integration and adaptive reward modeling, to further enhance performance in these difficult scenarios.
> > >
> > > We sincerely thank you again for your constructive feedback and support.

---

### Official Review · Reviewer_BcrF · 2025-07-20

**Clarity:** 3
**Significance:** 3
**Originality:** 3
**Rating:** 4
**Confidence:** 4

**Summary:**

This paper proposes SolverLLM, an LLM-guided MCTS framework for solving optimization problems. SolverLLM contains 4 phases: selection, dynamic expansion, simulation and backpropagation. The main new components in this work are: (1) dynamic expansion of non-leaf nodes in the search tree; (2) prompt backpropagation to incorporate rich non-scalar feedback; and (3) uncertainty backpropagation to deal with the high variance of LLM-based scoring. The authors evaluate their approach on 6 optimization benchmarks, and they demonstrate that SolverLLM consistently improves the performance over baseline methods.

**Questions:**

1. Please show scaling curves on how the model performance improves with more search budget, such as the number of search iterations, search nodes, etc.

2. Please add discussion on the token budget of SolverLLM compared to baselines, and how it grows with more search iterations.

3. It would be helpful if the authors can present a small example to illustrate how the full workflow of SolverLLM, including the concrete prompt in each stage, how does the solver and LLM-generated feedback look like, what is the output format of the LLM-based scorer and how it is used, etc.

4. I am confused about how prompt propagation works. What are the concrete formats of reasoning signals and the trigger, and how are they generated? How do we incorporate these reasoning signals for future prompt construction? Do we concatenate all past evaluation outcomes in the prompt?

5. I didn't find the prompt for generating objective scores. How does the prompt look like? What is the format of the objective score?

**Ethical Concerns:**

["NO or VERY MINOR ethics concerns only"]

**Final Justification:**

The author rebuttal addressed my questions. Therefore, I keep my review score.

**Limitations:**

Yes.

**Paper Formatting Concerns:**

No.

**Quality:**

3

**Strengths And Weaknesses:**

Strengths:

1. Using LLM for optimization problems is an important topic, and this work demonstrates strong results on diverse benchmarks.

2. The new design upon the standard MCTS framework is well-motivated and effective.

Weaknesses:

1. One main weakness of this work is the lack of a rigorous study of the inference cost of SolverLLM and other test-time scaling baselines. Specifically, the authors only demonstrate the final results and average generation time. However, there are no scaling curves to show how the model performance improves with more search budget, such as the number of search iterations, search nodes, etc. Therefore, it is unclear how much improvement SolverLLM provides when fixing the same search budget compared to baselines. It would be more informative to present such test-time scaling curves to compare different methods.

2. Related to the above question, there is no discussion on the token budget of SolverLLM compared to baselines, and how it grows with more search iterations. It is possible that SolverLLM uses a higher token budget in each search iteration compared to baselines.

3. Another major weakness is that the lack of clarity on method description. It would be helpful if the authors can present a small example to illustrate how SolverLLM works, including the concrete prompt in each stage, how does the solver and LLM-generated feedback look like, what is the output format of the LLM-based scorer and how is it used, etc.

4. I am confused about how prompt propagation works. What are the concrete formats of reasoning signals and the trigger, and how are they generated? How do we incorporate these reasoning signals for future prompt construction? Do we concatenate all past evaluation outcomes in the prompt?

5. I didn't find the prompt for generating objective scores. How does the prompt look like? What is the format of the objective score?

---

> ### Author Rebuttal · Authors · 2025-07-30
>
> We sincerely thank Reviewer BcrF for recognizing the relevance of our topic as well as the novelty and effectiveness of our proposed approach. The concerns raised mainly stem from the trade-off between inference cost and performance gains, as well as certain implementation details. We address these points with clarifications in the following response.
>
> > ### [W1 & Q1] Concern about the search budget.
>
> We agree that inference overhead and its relation to performance improvements are essential aspects when evaluating test-time scaling methods. To further address this, we conducted additional experiments analyzing the impact of search budget on performance. Specifically, on the MamoComplex dataset, we plotted curves of Solving Accuracy (SA) and Average Generation Time (AGT) as the maximum number of search iterations and the maximum number of components per expansion increase, as shown in the table below:
>
> | # Search Iterations | # Components per Expansion | SA (%) | AGT (min) |
> |---------------------|----------------------------|--------|-----------|
> | 10                  | 3                          | 69.0   | 2.54      |
> | 20                  | 3                          | 76.0   | 3.85      |
> | 30                  | 3                          | 77.0   | 4.82      |
> | 40                  | 3                          | 78.0   | 5.98      |
> | 50                  | 3                          | 78.0   | 6.73      |
> | 20                  | 1                          | 72.0   | 2.67      |
> | 20                  | 2                          | 74.0   | 3.24      |
> | 20                  | 4                          | 76.0   | 4.21      |
> | 20                  | 5                          | 76.0   | 4.88      |
>
> As shown, there is a clear trade-off between efficiency and effectiveness. We ultimately chose 20 search iterations and 3 components per expansion, as this setting achieves the best balance: fewer search budgets result in insufficient performance, while larger budgets increase AGT without significant performance gains.
>
> In the final version, we will include the scaling curves in the main text or appendix to more intuitively illustrate the test-time scaling characteristics of SolverLLM. We believe this addition will further demonstrate that our method achieves stronger performance than baselines while maintaining inference efficiency.
>
> > ### [W2 & Q2] Concern about the token budget.
>
> We recognize the importance of token budget in test-time scaling methods, as it directly affects both cost and feasibility in practical applications. To address this, we conducted supplementary experiments measuring the average token consumption of SolverLLM on the MamoComplex dataset under different numbers of search iterations, and compared the results with AutoFormulation, as shown below:
>
> | # Search Iterations | SolverLLM (# Tokens / SA %) | AutoFormulation (# Tokens / SA %) |
> |-------------------|---------------------------|---------------------------------|
> | 10                | 32,790 / 69.0            | 35,911 / 34.0                  |
> | 20                | 40,920 / 76.0            | 43,150 / 37.0                  |
> | 30                | 49,337 / 77.0            | 54,427 / 38.0                  |
> | 40                | 57,834 / 78.0            | 62,248 / 38.0                  |
> | 50                | 66,312 / 78.0            | 70,245 / 40.0                  |
>
> SolverLLM achieves better performance than AutoFormulation while using fewer tokens at the same number of search iterations due to a more efficient search strategy. In the final version, we will explicitly report the token consumption curves over varying search budgets to further enhance result transparency and comparability.
>
> > ### [W3-W5 & Q3-Q5] Concern about the detailed method implementation.
>
> We appreciate your thorough consideration. Questions W3–W5 all pertain to detailed aspects of SolverLLM, which we address collectively here. The specific prompts used at each search stage are included in Appendix C.4 for your reference. We model the optimization problems using Pyomo and solve them with concrete solvers such as CBC and IPOPT.
>
> LLM-generated feedback is a key component for understanding SolverLLM. After evaluating a complete formulation, the feedback takes the following form and is subsequently used during the backpropagation phase:
>
> ```json
> {
>   "score": "a value between 0 and 100",
>   "evaluation": {
>     "type": {
>       "need_revise": "1 (True) or 0 (False)",
>       "reason": "Justification for the 'need_revise' decision",
>       "prompt": "Prompt that helps avoid the same problem in future",
>       "uncertainty": "Predictive entropy computed from the 'reason' field"
>     },
>     "sets": ... (same structure as above),
>     "parameters": ...,
>     "variables": ...,
>     "objective": ...,
>     "constraints": ...
>   }
> }
> ```
>
> The objective score is obtained by repeatedly sampling according to the following prompt and averaging the results. The semantic uncertainty is treated as a global uncertainty that contributes to the reward calculation of each node in subsequent steps.
>
> Partial prompt for objective score (For W5 & Q5):
>
> ```
> The task is to evaluate this formulation according to the following criteria:
> 1. Correctness: Does the formulation correctly model the problem described?
> 2. Completeness: Does the formulation include all necessary components?
> 3. Efficiency: Is the formulation efficient in terms of variables and constraints?
> 4. Solvability: Did the execution produce a correct solution?
> 5. Solution quality: Does the solution make sense for the given problem?
>
> Based on your evaluation, provide a numerical score from 0 to 100, where:
> - 0-20: Poor formulation with major flaws
> - 21-40: Flawed formulation with significant issues
> - 41-60: Adequate formulation with some issues
> - 61-80: Good formulation with minor issues
> - 81-100: Excellent formulation that correctly models the problem
> ```
>
> Regarding question Q4, the `prompt` field in the feedback is used as local guidance for future prompt construction depending on the local uncertainty at each level and the `need_revise` flag (requiring uncertainty > threshold $\eta$ and `need_revise` = 1). During a complete MCTS process, all valid local guidances are used to guide generation results at that level. As for all past evaluation outcomes, they are incorporated into each node’s reward, so concatenating all past feedback is unnecessary.
>
> Again, we greatly appreciate the reviewer BcrF’s comments and hope our clarifications are helpful. We welcome continued dialogue to resolve any remaining concerns.

---

> ### Author Response · Authors · 2025-08-08
>
> We truly appreciate your earlier comments and the valuable insights you have provided. And we kindly follow up on our rebuttal and would be grateful for any further comments or clarifications you might wish to provide.

---

### Author Response · Authors · 2025-08-08
**Thank you and author-reviewer discussion**

We appreciate that the reviewers generally have positive impressions of our work, including

**Motivation**
+ "Using LLM for optimization problems is an important topic"

**Novelty**
+ "The new design upon the standard MCTS framework is well-motivated and effective"
+ "The inclusion of type element is sensible, serving as a general guideline to narrow down the possible modeling choices for LLMs"
+ "The use of active nodes allows dynamically retrying at the specific levels, which is also a sensible design choice"
+ "The integration of knowledge-base for each level provides insights for revising incorrect formulations"
+ "Integrates MCTS with three novel feedback mechanisms that demonstrably boost solving accuracy without any extra training"

**Evaluation**
+ "this work demonstrates strong results on diverse benchmarks"
+ "Demonstrates consistent accuracy gains over existing prompt-based methods on benchmark datasets"
+ "Achieves up to 10 pp absolute accuracy gains over state-of-the-art prompt- and learning-based baselines while preserving near-100 % executable code"

During the rebuttal period, the reviewers raised many valuable comments. Among these, the most frequently mentioned concerns and our efforts to address them are as follows:

> Concern regarding the significance and novelty.

We highlight the significance and novelty of SolverLLM from three perspectives:

1. Unique Research Challenge: Automatically translating natural language into executable, high-quality optimization formulations is a complex and long-standing open problem that goes beyond direct solution generation, requiring structured reasoning, cross-task generalization, and robust error handling.
2. Methodological Innovations: SolverLLM introduces a novel combination of dynamic expansion, prompt backpropagation, and uncertainty propagation within a test-time reasoning framework. These components enhance search flexibility, leverage solver feedback for guided generation, and improve inference stability by modeling uncertainty—advancing beyond prior MCTS-LLM approaches.
3. Paradigm Shift: Rather than incremental changes to vanilla MCTS, SolverLLM proposes a training-free, unsupervised framework tightly integrated with LLMs, enabling cross-domain optimization without fine-tuning or labeled data. This introduces a new paradigm for applying LLMs to complex decision-making tasks.

> Concern regarding the resource consumption.

We conducted two additional experiments analyzing the impact of search budget on performance and measuring the average token consumption of SolverLLM under different numbers of search iterations, comparing the results with AutoFormulation. The results show that (1) fewer search budgets result in insufficient performance, while larger budgets increase average generation time (AGT) without significant performance gains; (2) SolverLLM achieves better performance than AutoFormulation while using fewer tokens at the same number of search iterations due to a more efficient search strategy.

Thank you again to all reviewers for your effort and time. We would appreciate it if you could review our response and let us know if you have any further questions. We look forward to your feedback!

---

### Note · Authors · 2025-08-13

We sincerely appreciate the constructive comments from all reviewers, and summarize main contributions, concerns, and responses.

**Main Contributions:**

We propose SolverLLM, a training-free framework that leverages test-time scaling to solve diverse optimization problems by three novel and clever mechanisms through a MCTS process; extensive experiments on six datasets show that SolverLLM has achieved state-of-the-art performance.

**Main Concerns and Responses:**

**Reviewer BcrF:**

> Lack of a rigorous study of the inference cost and token budget.

We provided new experiments showing SolverLLM’s clear efficiency–effectiveness trade-off and its token consumption.

> Lack of clarity on method description.

We conducted additional clarifications by providing concrete prompt examples and detailed feedback structure.

**Reviewer SNpT:**

> No solver-algorithm contribution.

We emphasized our novelty again and clarified that solver algorithm contribution is not our main focus.

> No comparison on solver time cost with human experts.

We explained that SolverLLM targets accessibility and low human/training cost, with extensive comparisons to diverse baselines.

**Reviewer 2ytH:**

> The significance and novelty seem limited.

We once again emphasized our novelty from multiple aspects.

> Prompt backpropagation is somewhat misleading.

We provided a more detailed explanation of the Prompt Backpropagation mechanism to address the reviewer's concerns.

> The experiments are not fully clear.

We provided sensitivity analysis of lambda and empirical estimation of token consumption, and further analyzed them.

**Reviewer 6kR9:**

> Lack of test-time latency and resource consumption analysis.

We conducted an analysis through additional experiments.

> The MCTS procedure need clearer explanations.

We have conducted an analysis of the entire MCTS process based on the reviewer's questions.

> The design of uncertainty back-propagation is somewhat confusing.

We provided a detailed explanation of each uncertainty used in the method and explained their connections, demonstrating their necessity.

We appreciate that most reviewers, indluding BcrF, SNpT, and 2ytH, expressed generally positive impressions of our work. As revewer BcrF and 6kR9 has not explicitly indicated whether the concerns are fully resolved, we would greatly appreciate it if you could kindly encourage further discussion in your private discussion section.

Best regards.

---

### Decision · Program_Chairs · 2025-09-17

**Decision:**

Accept (poster)

**Comment:**

This paper introduces SolverLLM, a training free framework leveraging LLM guided MCTS to generate solver-ready (e.g. Gurobi, Pyomo) optimization formulations from natural language descriptions at test time. The method introduces 1) dynamic expansion for incremental formulation construction - producing new open ended and context aware child nodes, 2) prompt backpropagation for solver feedback guided refinement, and 3) uncertainty backpropagation for improving search robustness. Experiments on six optimization datasets show consistent improvement over prompt based and test time scaling baselines.

Strengths:
The paper addresses an important problem of converting natural language descriptions to solver ready optimization formulations. The experiments show significant gains across diverse benchmarks outperforming strong baselines such as AutoFormulation.

Weaknesses:
Several reviewers asked for more systemic test time complexity, scaling curves, solver time cost, runtime statistics, and token cost comparisons

Some connections to prior work (Reflexion, LATS) should be clarified to highlight the relative contribution - reviewer `6kR9'

Clarify of exposition regarding MCTS procedure and relationship between node granularity, six-element schema (Type, Sets, Parameters, Variables, Objective, and Constraints in section 3.2.1) could be improved for readability.

The authors provided additional experiments on scaling behavior, token consumption which addressed many concerns raised by the reviewers. Several reviewers increased their scores after these clarifications. While some reservations remain about complexity and broader applicability, the consensus was that the contributions are sufficiently novel and the empirical results are significant.

The decision is to accept. I strongly encourage the authors to incorporate the feedback - include the scaling and token-budget analyses (every new experiments appeared in the rebuttal as per reviewer `BcrF`), clarify the MCTS details in section 3.2.1, distinct role of each uncertainty mechanisms, and clarify the contribution with respect to the related approaches.